# Test-Time Adaptation for Visual Document Understanding

**Sayna Ebrahimi**                                                    *saynae@google.com*
*Google Cloud AI Research*

**Sercan Ö. Arik**                                                    *soarik@google.com*
*Google Cloud AI Research*

**Tomas Pfister**                                                     *tpfister@google.com*
*Google Cloud AI Research*

**Reviewed on OpenReview:** *https://openreview.net/forum?id=zshemTAa6U*

## Abstract

For visual document understanding (VDU), self-supervised pretraining has been shown to successfully generate transferable representations, yet, effective adaptation of such representations to distribution shifts at test-time remains to be an unexplored area. We propose DocTTA, a novel test-time adaptation method for documents, that does source-free domain adaptation using unlabeled target document data. DocTTA leverages cross-modality self-supervised learning via masked visual language modeling, as well as pseudo labeling to adapt models learned on a *source* domain to an unlabeled *target* domain at test time. We introduce new benchmarks using existing public datasets for various VDU tasks, including entity recognition, key-value extraction, and document visual question answering. DocTTA shows significant improvements on these compared to the source model performance, up to 1.89% in (F1 score), 3.43% (F1 score), and 17.68% (ANLS score), respectively. Our benchmark datasets are available at `https://saynaebrahimi.github.io/DocTTA.html`.

## 1 Introduction

Visual document understanding (VDU) is on extracting structured information from document pages represented in various visual formats. It has a wide range of applications, including tax/invoice/mortgage/claims processing, identity/risk/vaccine verification, medical records understanding, compliance management, etc. These applications affect operations of businesses from major industries and daily lives of the general populace. Overall, it is estimated that there are trillions of documents in the world.

Machine learning solutions for VDU should rely on overall comprehension of the document content, extracting the information from text, image, and layout modalities. Most VDU tasks including key-value extraction, form understanding, document visual question answering (VQA) are often tackled by self-supervised pretraining, followed by supervised fine-tuning using human-labeled data (Appalaraju et al., 2021; Gu et al., 2021; Xu et al., 2020b;a; Lee et al., 2022; Huang et al., 2022). This paradigm uses unlabeled data in a task-agnostic way during the pretraining stage and aims to achieve better generalization at various downstream tasks. However, once the pretrained model is fine-tuned with labeled data on *source* domain, a significant performance drop might occur if these models are directly applied to a new unseen *target* domain – a phenomenon known as *domain shift* (Quiñonero-Candela et al., 2008a;b; Moreno-Torres et al., 2012).

The domain shift problem is commonly encountered in real-world VDU scenarios where the training and test-time distributions are different, a common situation due to the tremendous diversity observed for document data. Fig. 1 exemplifies this, for key-value extraction task across visually different document templates and for visual question answering task on documents with different contents (figures, tables, letters etc.) for information. The performance difference due to this domain shift might reduce the stability and reliability of VDU models. This is highly undesirable for widespread adoption of VDU, especially given that the common

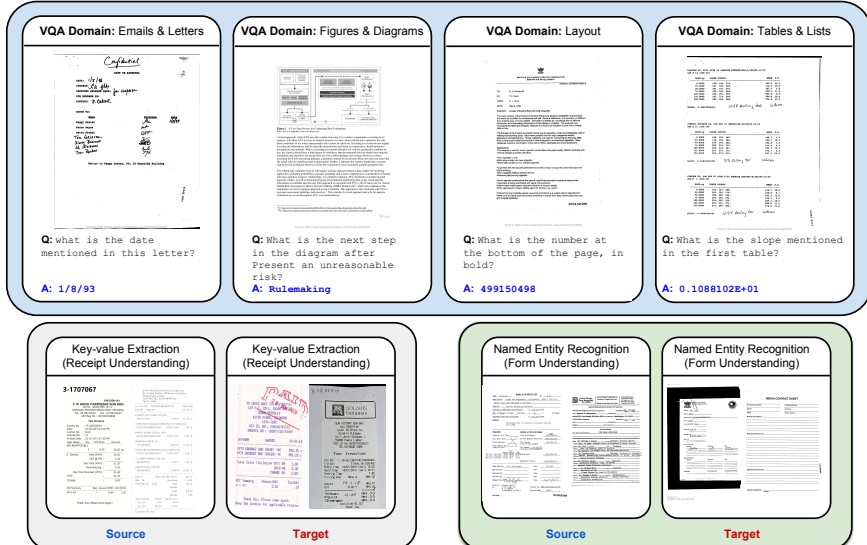

Figure 1: Distribution shift examples for document samples from the proposed benchmark, DocVQA-TTA. **Top row:** shows documents from four domains: (i) Emails & Letters, (ii) Figures & Diagrams, (iii) Layout, (iv) Tables & Lists, from our VQA benchmark derived from DocVQA dataset (Mathew et al., 2021). **Bottom left:** documents from source and target domains for key-value information extraction task from SROIE (Huang et al., 2019) receipt dataset. **Bottom right:** documents from source and target domains for named entity recognition task from FUNSD (Jaume et al., 2019) dataset.

use cases are from high-stakes applications from finance, insurance, healthcare, or legal. Thus, the methods to robustly guarantee high accuracy in the presence of distribution shifts would be of significant impact. Despite being a critical issue, to the best of our knowledge, no prior work has studied post-training domain adaptation for VDU.

Unsupervised domain adaptation (UDA) methods attempt to mitigate the adverse effect of data shifts, often by training a joint model on the labeled source and unlabeled target domains that map both domains into a common feature space. However, simultaneous access to data from source and target domains may not be feasible for VDU due to the concerns associated with source data access, that might arise from legal, technical, and contractual privacy constraints. In addition, the training and serving may be done in different computational environments, and thus, the expensive computational resources used for training may not be available at serving time. Test-time adaptation (TTA) (or source-free domain adaptation) has been introduced to adapt a model that is trained on the source to unseen target data, without using any source data (Liang et al., 2020; Wang et al., 2021b; Sun et al., 2020; Wang et al., 2021a; Chen et al., 2022; Huang et al., 2021). TTA methods thus far have mainly focused on image classification and semantic segmentation tasks, while VDU remains unexplored, despite the clear motivations of the distribution shift besides challenges for the employment of standard UDA.

Since VDU significantly differs from other computer vision (CV) tasks, applying existing TTA methods in a straightforward manner is suboptimal. First, for VDU, information is extracted from multiple modalities (including image, text, and layout) unlike other CV tasks. Therefore, a TTA approach proposed for VDU should leverage cross-modal information for better adaptation. Second, multiple outputs (e.g. entities or questions) are obtained from the same document, creating the scenario that their similarity in some aspects (e.g. in format or context) can be used. However, this may not be utilized in a beneficial way with direct application of popular pseudo labeling or self training-based TTA approaches (Lee et al., 2013), which have gained a lot of attention in CV (Liang et al., 2020; 2021; Chen et al., 2022; Wang et al., 2021a). Pseudo labeling uses predictions on unlabeled target data for training. However, naive pseudo labeling can result in accumulation of errors (Guo et al., 2017; Wei et al., 2022; Meinke & Hein, 2020; Thulasidasan et al., 2019; Kristiadi et al., 2020). Particularly in VDU, it happens due to generation of multiple outputs at the same time that are possibly wrong in the beginning, as each sample can contain a long sequence of

words. Third, commonly-used self-supervised contrastive-based TTA methods for CV (He et al., 2020; Chen et al., 2020b;a; Tian et al., 2020) (that are known to improve generalization) employ a rich set of image augmentation techniques, while proposing data augmentation is much more challenging for general VDU.

In this paper, we propose DocTTA, a novel TTA method for VDU that utilizes self-supervised learning on text and layout modalities using masked visual language modeling (MVLM) while jointly optimizing with pseudo labeling. We introduce an uncertainty-aware per-batch pseudo labeling selection mechanism, which makes more accurate predictions compared to the commonly-used pseudo labeling techniques in CV that use no pseudo-labeling selection mechanism (Liang et al., 2020) in TTA or select pseudo labels based on both uncertainty and confidence (Rizve et al., 2021) in semi-supervised learning settings. To the best of our knowledge, this is the first method with a self-supervised objective function that combines visual and language representation learning as a key differentiating factor compared to TTA methods proposed for image or text data. While our main focus is the TTA setting, we also showcase a special form of DocTTA where access to source data is granted at test time, extending our approach to be applicable for unsupervised domain adaptation, named DocUDA. Moreover, in order to evaluate DocTTA diligently and facilitate future research in this direction, we introduce new benchmarks for various VDU tasks including key-value extraction, entity recognition, and document visual question answering (DocVQA) using publicly available datasets, by modifying them to mimic real-world adaptation scenarios. We show DocTTA significantly improves source model performance at test-time on all VDU tasks without any supervision. To our knowledge, our paper is first to demonstrate TTA and UDA for VDU, showing significant accuracy gain potential via adaptation. We expect our work to open new horizons for future research in VDU and real-world deployment in corresponding applications.

## 2 Related work

**Unsupervised domain adaptation** aims to improve the performance on a different target domain, for a model trained on the source domain. UDA approaches for closed-set adaptation (where classes fully overlap between the source and target domains) can be categorized into four categories: (i) distribution alignment-based, (ii) reconstruction-based, and (iii) adversarial based, and (iv) pseudo-labeling based. Distribution alignment-based approaches feature aligning mechanisms, such as moment matching (Peng et al., 2019) or maximum mean discrepancy (Long et al., 2015; Tzeng et al., 2014). Reconstruction-based approaches reconstruct source and target data with a shared encoder while performing supervised classification on labeled data (Ghifary et al., 2016), or use cycle consistency to further improve domain-specific reconstruction (Murez et al., 2018; Hoffman et al., 2018). Inspired by GANs, adversarial learning based UDA approaches use two-player games to disentangle domain invariant and domain specific features (Ganin & Lempitsky, 2015; Long et al., 2018; Shu et al., 2018). Pseudo-labeling (or self-training) approaches jointly optimize a model on the labeled source and pseudo-labeled target domains for adaptation (Kumar et al., 2020; Liu et al., 2021; French et al., 2017). Overall, all UDA approaches need to access both labeled source data and unlabeled target data during the adaptation which is a special case for the more challenging setting of TTA and we show how our approach can be modified to be used for UDA.

**Test-time adaptation** corresponds to source-free domain adaptation, that focuses on the more challenging setting where only source model and unlabeled target data are available. The methods often employ an unsupervised or self-supervised cost function. Nado et al. (2020) adopted the training time BN and used it at test time to adapt to test data efficiently. In standard practice, the batch normalization statistics are frozen to particular values after training time that are used to compute predictions. Nado et al. (2020) proposed to recalculate batch norm statistics on every batch at test time instead of re-using values from training time as a mechanism to adapt to unlabeled test data. While this is computationally inexpensive and widely adoptable to different settings, we empirically show that it cannot fully overcome performance degradation issue when there is a large distribution mismatch between source and target domains. TENT (Wang et al., 2021b) utilizes entropy minimization for test-time adaptation encouraging the model to become more "certain" on target predictions regardless of their correctness. In the beginning of the training when predictions tend to be inaccurate, entropy minimization can lead to error accumulation since VDU models create a long sequence of outputs per every document resulting in a noisy training. SHOT (Liang et al.,

2020) combines mutual information maximization with offline clustering-based pseudo labeling. However, using simple offline pseudo-labeling can lead to noisy training and poor performance when the distribution shifts are large (Chen et al., 2022; Liu et al., 2021; Rizve et al., 2021; Mukherjee & Awadallah, 2020). We also use pseudo labeling in DocTTA but we propose online updates per batch for pseudo labels, as the model adapts to test data. Besides, we equip our method with a pseudo label rejection mechanism using uncertainty, to ensure the negative effects of predictions that are likely to be inaccurate. Most recent TTA approaches in image classification use contrastive learning combined with extra supervision (Xia et al., 2021; Huang et al., 2021; Wang et al., 2021a; Chen et al., 2022). In contrastive learning, the idea is to jointly maximize the similarity between representations of augmented views of the same image, while minimizing the similarity between representations of other samples). All these methods rely on self-supervised learning that utilize data augmentation techniques, popular in CV while not yet being as effective for VDU. While we advocate for using SSL during TTA, we propose to employ multimodal SSL with pseudo labeling for the first time which is imore effective for VDU.

**Self-supervised pretraining for VDU** aims to learn generalizable representations on large scale unlabeled data to improve downstream VDU accuracy (Appalaraju et al., 2021; Gu et al., 2021; Xu et al., 2020b;a; Lee et al., 2022; Huang et al., 2022). LayoutLM (Xu et al., 2020b) jointly models interactions between text and layout information using a masked visual-language modeling objective and performs supervised multi-label document classification on IIT-CDIP dataset (Lewis et al., 2006). LayoutLMv2 (Xu et al., 2020a) extends it by training on image modality as well, and optimizing text-image alignment and text-image matching objective functions. DocFormer (Appalaraju et al., 2021) is another multi-modal transformer based architecture that uses text, vision and spatial features and combines them using multi-modal self-attention with a multi-modal masked language modeling (MM-MLM) objective (as a modified version of MLM in BERT (Devlin et al., 2018)), an image reconstruction loss, and a text describing image loss represented as a binary cross-entropy to predict if the cut-out text and image are paired. FormNet (Lee et al., 2022) is a structure-aware sequence model that combines a transformer with graph convolutions and proposes *rich attention* that uses spatial relationship between tokens. UniDoc (Gu et al., 2021) is another multi-modal transformer based pretraining method that uses masked sentence modeling, visual contrastive learning, and visual language alignment objectives which unlike other methods, does not have a fixed document object detector (Li et al., 2021; Xu et al., 2020a). In this work, we focus on a novel TTA approach for VDU, that can be integrated with any pre-training method. We demonstrate DocTTA using the publicly available LayoutLMv2 architecture pretrained on IIT-CDIP dataset.

## 3   DocTTA: Test-time adaptation for documents

In this section, we introduce DocTTA, a test-time adaptation framework for VDU tasks including key-value extraction, entity recognition, and document visual question answering (VQA).

### 3.1   DocTTA framework

We define a *domain* as a pair of distribution $\mathcal{D}$ on inputs $\mathcal{X}$ and a labeling function $l : \mathcal{X} \to \mathcal{Y}$. We consider *source* and *target* domains. In the source domain, denoted as $\langle D_s, l_s \rangle$, we assume to have a model denoted as $f_s$ and parameterized with $\theta_s$ to be trained on source data $\{x_s^{(i)}, y_s^{(i)}\}_{i=1}^{n_s}$, where $x_s^{(i)} \in \mathcal{X}_s$ and $y_s^{(i)} \in \mathcal{Y}_s$ are document inputs and corresponding labels, respectively and $n_s$ is the number of documents in the source domain. Given the trained source model $f_s$ and leaving $\mathcal{X}_s$ behind, the goal of TTA is to train $f_t$ on the target domain denoted as $\langle \mathcal{D}_t, l_t \rangle$ where $f_t$ is parameterized with $\theta_t$ and is initialized with $\theta_s$ and $\mathcal{D}_t$ is defined over $\{x_t^{(i)}\}_{i=1}^{n_t} \in \mathcal{X}_t$ without any ground truth label. Algorithm 1 overviews our proposed DocTTA procedure.

Unlike single-modality inputs commonly used in computer vision, documents are images with rich textual information. To extract the text from the image, we consider optical character recognition (OCR) is performed and use its outputs, characters, and their corresponding bounding boxes (details are provided in Appendix). We construct our input $\mathcal{X}$ in either of the domains composed of three components: text input sequence $X^T$ of length $n$ denoted as $(x_1^T, \cdots, x_n^T) \in \mathbb{R}^{(n \times d)}$, image $X^I \in \mathbb{R}^{3 \times W \times H}$, and layout $X^B$ as a 6-dimensional

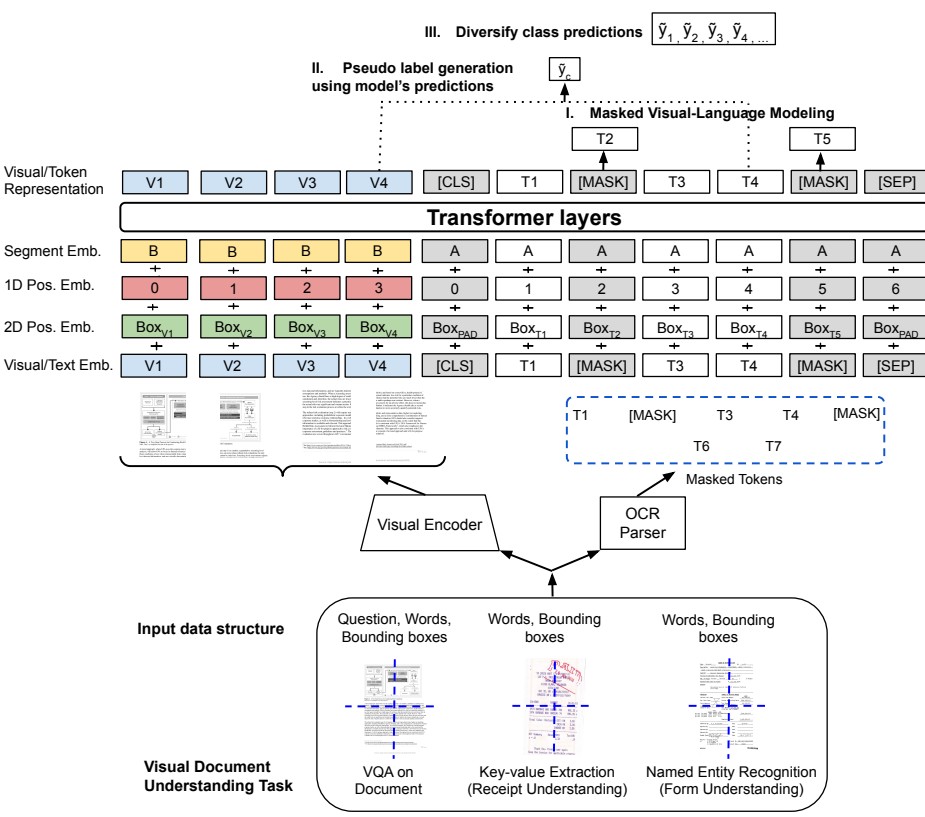

Figure 2: Illustration of how our approach, DocTTA, leverages unlabeled target data at test time to i) learn how to predict masked language given visual cues, ii) generate pseudo labels to supervise the learning, and iii) maximize the diversity of predictions to generate sufficient amount labels from all classes.

vector in the form of $(x_{min}, x_{max}, y_{min}, y_{max}, w, h)$ representing a bounding box associated with each word in the text input sequence. Note that for the VQA task, text input sequence is also prepended with the question. For the entity recognition task, labels correspond to the set of classes that denote the extracted text; for the key-value extraction task, labels are values for predefined keys; and for the VQA task, labels are the starting and ending positions of the answer presented in the document for the given question. We consider the closed-set assumption: the source and target domains share the same class labels $\mathcal{Y}_s = \mathcal{Y}_t = \mathcal{Y}$ with $|\mathcal{Y}| = C$ being the total number of classes.

---

**Algorithm 1** DocTTA for closed-set TTA in VDU

---

1: **Input:** Source model weights $\theta_s$, target documents $\{x_t^i\}_{i=1}^{n_t}$, test-time training epochs $n_e$, test-time training learning rate $\alpha$, uncertainty threshold $\gamma$, questions for target documents in document VQA task

2: **Initialization:** Initialize target model $f_{\theta_t}$ with $\theta_s$ weights.
3: **for** $epoch = 1$ to $n_e$ **do**
4:    Perform masked visual-language modeling in Eq. 1
5:    Generate pseudo labels and accept a subset using criteria in Eq. 2 and fine-tune with Eq. 3
6:    Maximize diversity in pseudo label predictions Eq. 4
7:    $\theta_t \leftarrow \theta_t - \alpha\nabla\mathcal{L}_{\text{DocTTA}}$                          ▷ Update $\theta_t$ via total loss in Eq. 5
8: **end for**

---

## 3.2 DocTTA objective functions

In order to adapt $f_t$ in DocTTA, we propose three objectives to optimize on the unlabeled target data:

**Objective I: masked visual language modeling (MVLM).** Inspired by notable success of masked language modeling for NLP in architectures like BERT (Devlin et al., 2018), as well as the success of MVLM in (Xu et al., 2020a) to perform self-supervised pretraining for VDU, we propose to employ MVLM at test time to encourage the model to learn better the text representation of the test data given the 2D positions and other text tokens. The intuition behind using this objective for TTA is to enable the target model to learn the language modality of the new data given visual cues and thereby bridging the gap between the different modalities on the target domain. We randomly mask 15% of input text tokens among which 80% are replaced by a special token [MASK] and the remaining tokens are replaced by a random word from the entire vocabulary. The model is then trained to recover the masked tokens while the layout information remains fixed. To do so, the output representations of masked tokens from the encoder are fed into a classifier which outputs logits over the whole vocabulary, to minimize the negative log-likelihood of correctly recovering masked text tokens $x_m^T$ given masked image tokens $x^I$ and masked layout $x^B$:

$$\mathcal{L}_{MVLM}(\theta_t) = -\mathbb{E}_{x_t \in \mathcal{X}_t} \sum_m \log p_{\theta_t}(x_{t_m}^T | x_t^I, x_t^B). \tag{1}$$

**Objective II: self training with pseudo labels.** While optimizing MVLM loss during the adaptation, we also generate pseudo labels for the unlabeled target data in an online way and treat them as ground truth labels to perform supervised learning on the target domain. Unlike previous pseudo labeling-based approaches for TTA in image classification which update pseudo labels only after each epoch (Liang et al., 2020; 2021; Wang et al., 2021a), we generate pseudo labels per batch aiming to use the latest version of the model for predictions. We consider a full epoch to be one training loop where we iterate over the entire dataset batch by batch. In addition, unlike prior works, we do not use a clustering mechanism to generate pseudo labels as they will be computationally expensive for documents. Instead, we directly use predictions by the model. However, simply using all the predictions would lead to noisy pseudo labels.

Inspired by (Rizve et al., 2021), in order to prevent noisy pseudo labels, we employ an uncertainty-aware selection mechanism to select the subset of pseudo labels with low uncertainty. Note that in (Rizve et al., 2021), pseudo labeling is used as a semi-supervised learning approach (not adaptation) and the selection criteria is based on both thresholding confidence and uncertainty where confidence is the Softmax probability and uncertainty is measured with MC-Dropout (Gal & Ghahramani, 2016). We empirically observe that raw confidence values (when taken as the posterior probability output from the model) are overconfident despite being right or wrong. Setting a threshold on pseudo labels' confidence only introduces a new hyperparameter without a performance gain (see Sec. 5.1). Instead, to select the predictions we propose to only use uncertainty, in the form of Shannon's entropy (Shannon, 2001). We also expect this selection mechanism leads to reducing miscalibration due to the direct relationship between the ECE[1] and output prediction uncertainty, i.e. when more certain predictions are selected, ECE is expected to reduce for the selected subset of pseudo labels. Assume $\mathbf{p}^{(i)}$ be the output probability vector of the target sample $x_t^{(i)}$ such that $p_c^{(i)}$ denotes the probability of class $c$ being the correct class. We select a pseudo label $\tilde{y}_c^{(i)}$ for $x_t^{(i)}$ if the uncertainty of the prediction $u(p_c^{(i)})$, measured with Shannon's entropy, is below a specific threshold $\gamma$ and we update $\theta_t$ weights with a cross-entropy loss:

$$\tilde{y}_c^i = \mathbb{1}\left[u(p_c^{(i)}) \leq \gamma\right], \tag{2}$$

$$\mathcal{L}_{CE}(\theta_t) = -\mathbb{E}_{x_t \in \mathcal{X}_t} \sum_{c=1}^C \tilde{y}_c \log \sigma(f_t(x_t)), \tag{3}$$

where $\sigma(\cdot)$ is the softmax function. It should be noted that the tokens that are masked for the MVLM loss are not included in the cross-entropy loss as the attention mask for them is zero.

**Objective III: diversity objective.** To prevent the model from indiscriminately being dominated by the most probable class based on pseudo labels, we encourage class diversification in predictions by minimizing

---

[1]Expected calibration error (Naeini et al., 2015) which is a metric to measure calibration of a model

the following objective:

$$\mathcal{L}_{DIV} = \mathbb{E}_{x_t \in \mathcal{X}_t} \sum_{c=1}^{C} \bar{p}_c \log \bar{p}_c, \tag{4}$$

where $\bar{p} = \mathbb{E}_{x_t \in \mathcal{X}_t} \sigma(f_t(x_t))$ is the output embedding of the target model averaged over target data. By combining Eqs. 1, 3, and 4, we obtain the full objective function in DocTTA as below:

$$\mathcal{L}_{\text{DocTTA}} = \mathcal{L}_{MVLM} + \mathcal{L}_{CE} + \mathcal{L}_{DIV}. \tag{5}$$

### 3.3 DocTTA vs. DocUDA

The proposed DocTTA framework can be extended as an UDA approach, which we refer to as DocUDA (see Appendix for the algorithm and details). DocUDA is based on enabling access to source data during adaptation to the target. In principle, the availability of this extra information of source data can provide an advantage over TTA, however, as we show in our experiments, their difference is small in most cases, and even TTA can be superior when source domain is significantly smaller than the target domain and the distribution gap is large, highlighting the efficacy of our DocTTA approach for adapting without relying on already-seen source data.

We note that the UDA version comes with fundamental drawbacks. From the privacy perspective, there would be concerns associated with accessing or storing source data in deployment environments, especially given that VDU applications are often from privacy-sensitive domains like legal or finance. From the computational perspective, UDA would yield longer convergence time and higher memory requirements due to joint learning from source data. Especially given that the state-of-the-art VDU models are large in size, this may become a major consideration.

## 4 DocTTA benchmarks

To better highlight the impact of distribution shifts and to study the methods that are robust against them, we introduce new benchmarks for VDU. Our benchmark datasets are constructed from existing popular and publicly-available VDU data to mimic real-world challenges. We have attached the training and test splits for all our benchmark datasets in the supplementary materials.

### 4.1 FUNSD-TTA: Entity recognition adaptation benchmark

We consider FUNSD, Form Understanding in Noisy Scanned Documents, dataset (Jaume et al., 2019) for this benchmark which is a noisy form understanding collection consists of sparsely-filled forms, with sparsity varying across the use cases the forms are from. In addition, the scanned images are noisy with different degradation amounts due to the disparity in scanning processes, which can further exacerbate the sparsity issue as the limited information might be based on incorrect OCR outputs. As a representative distribution shift challenge on FUNSD, we split the source and target documents based on the sparsity of available information measure. The original dataset has 9707 semantic entities and 31,485 words with 4 categories of entities `question`, `answer`, `header`, and `other`, where each category (except `other`) is either the beginning or the intermediate word of a sentence. Therefore, in total, we have 7 classes. We first combine the original training and test splits and then manually divide them into two groups. We set aside 149 forms that are filled with more texts for the source domain and put 50 forms that are sparsely filled for the target domain. We randomly choose 10 out of 149 documents for validation, and the remaining 139 for training. Fig. 1 (bottom row on the right) shows examples from the source and target domains.

### 4.2 SROIE-TTA: Key-value extraction adaptation benchmark

We use SROIE (Huang et al., 2019) dataset with 9 classes in total. Similar to FUNSD, we first combine the original training and test splits. Then, we manually divide them into two groups based on their visual appearance – source domain with 600 documents contains standard-looking receipts with proper angle of view and clear black ink color. We use 37 documents from this split for *validation*, which we use to tune

adaptation hyperparameters. Note that the validation split does not overlap with the target domain, which has 347 receipts with slightly blurry look, rotated view, colored ink, and large empty margins. Fig. 1 (bottom row on the left) exemplifies documents from the source and target domains.

### 4.3 DocVQA-TTA: Document VQA adaptation benchmark

We use DocVQA (Mathew et al., 2021), a large-scale VQA dataset with nearly 20 different types of documents including scientific reports, letters, notes, invoices, publications, tables, etc. The original training and validation splits contain questions from all of these document types. However, for the purpose of creating an adaptation benchmark, we select 4 *domains* of documents: i) *Emails & Letters* (**E**), ii) *Tables & Lists* (**T**), iii) *Figure & Diagrams* (**F**), and iv) *Layout* (**L**). Since DocVQA doesn't have public meta-data to easily sort all documents with their questions, we use a simple keyword search to find our desired categories of questions and their matching documents. We use the same words in domains' names to search among questions (i.e., we search for the words of "email" and "letter" for *Emails & Letters* domain). However, for *Layout* domain, our list of keywords is ["top", "bottom", "right", "left", "header", "page number"] which identifies questions that are querying information from a specific location in the document. Among the four domains, **L** and **E** have the shortest gap because emails/letters have structured layouts and extracting information from them requires understanding relational positions. For example, the name and signature of the sender usually appear at the bottom, while the date usually appears at top left. However, **F** and **T** domains seem to have larger gaps with other domains, that we attributed to that learning to answer questions on figures or tables requires understanding local information withing the list or table. Fig. 1 (top row) exemplifies some documents with their questions from each domain. Document counts in each domain are provided in Appendix.

## 5 Experiments

**Evaluation metrics:** For entity recognition and key-value extraction tasks, we use entity-level F1 score as the evaluation metric, whereas for the document VQA task, we use Average Normalized Levenshtein Similarity (ANLS) introduced by (Biten et al., 2019) (since it is recognized as a better measure compared to accuracy since it doesn't penalize minor text mismatches due to OCR errors).

**Model architecture:** In all of our experiments, we use LayoutLMv2$_{BASE}$ architecture which has a 12-layer 12-head Transformer encoder with a hidden size of 768. Its visual backbone is based on ResNeXt101-FPN, similar to that of MaskRCNN (He et al., 2017). Overall, it has ∼200M parameters. We note that our approach is architecture agnostic, and hence applicable to any attention-based VDU model. Details on training and hyper parameter tuning are provided in Appendix.

**Baselines:** As our method is the first TTA approach proposed for VDU tasks, there is no baseline to compare directly. Thus, we adopt TTA and UDA approaches from image classification, as they can be applicable for VDU given that they do not depend on augmentation techniques, contrastive learning, or generative modeling. For UDA, we use two established and commonly used baselines **DANN** (Ganin & Lempitsky, 2015) and **CDAN** (Long et al., 2018) and for TTA, we use the baselines batch normalization **BN** (Ioffe & Szegedy, 2015; Nado et al., 2020), **TENT** (Wang et al., 2021b), and **SHOT** (Liang et al., 2020). Details on all the baselines are given in Appendix A.2.2. We also provide **source-only**, where the model trained on source and evaluated on target without any adaptation mechanism, and **train-on-target**, where the model is trained (and tested) on target domain using the exact same hyperparameters used for TTA (which are found on the validation set and might not be the most optimal values for target data). While these two baselines don't adhere to any domain adaptation setting, they can be regarded as the ultimate lower and upper bound for performance.

### 5.1 Results and Discussions

**FUNSD-TTA:** Table 1 shows the comparison between DocTTA and DocUDA with their corresponding TTA and UDA baselines. For UDA, DocUDA outperforms all other UDA baselines by a large margin, and

Table 1: F1 score results for adapting source to target in **FUNSD-TTA** and **SROIE-TTA** benchmarks. Availability of the labeled/unlabeled data from source/target domains during *adaptation* in UDA and TTA settings and *training* phase in source-only and train-on-target settings are marked. Source-only and train-on-target serve as lower and upper bounds, respectively for performance on target domain. Standard deviations are in parentheses.

| DA category | Methods | Labeled source data | Labeled target data | Unlabeled target data | FUNSD-TTA | SROIE-TTA |
|---|---|:---:|:---:|:---:|---|---|
| - | source-only | ✓ | ✗ | ✗ | 80.80 (0.12) | 92.45 (0.08) |
| UDA | DANN | ✓ | ✗ | ✓ | 82.54 (0.14) | 92.89 (0.13) |
| | CDAN | ✓ | ✗ | ✓ | 83.72 (0.61) | 93.36 (0.18) |
| | **DocUDA** (ours) | ✓ | ✗ | ✓ | **89.76** (0.09) | **97.38** (0.15) |
| TTA | BN | ✗ | ✗ | ✓ | 80.84 (0.93) | 92.41 (0.45) |
| | TENT | ✗ | ✗ | ✓ | 79.78 (1.28) | 92.42 (0.87) |
| | SHOT | ✗ | ✗ | ✓ | 80.89 (1.03) | 92.78 (0.65) |
| | **DocTTA** (ours) | ✗ | ✗ | ✓ | **84.23** (0.88) | **94.34** (0.43) |
| - | train-on-target | ✗ | ✓ | ✗ | 99.89 (0.05) | 100.0 (0.00) |

improves 8.96% over the source-only. For the more challenging setting of TTA, DocTTA improves the F1 score of the source-only model by 3.43%, whereas the performance gain by all the TTA baselines is less than 0.5%. We also observe that DocTTA performs slightly better than other UDA baselines DANN and CDAN, which is remarkable given that unlike those, DocTTA does not have access to the source data at test time.

**SROIE-TTA:** Table 1 shows the comparison between UDA and TTA baselines vs. DocUDA and DocTTA on SROIE-TTA benchmark. Similar to our findings for FUNSD-TTA, DocUDA and DocTTA outperform their corresponding baselines, where DocTTA can even surpass DANN and CDAN (which use source data at test time). Comparison of DocUDA and DocTTA shows that for small distribution shifts, UDA version of our framework results in better performance.

**DocVQA-TTA:** Table 2 shows the results on our DocVQA-TTA benchmark, where the ANLS scores are obtained by adapting each domain to all the remaining ones. The distribution gap between domains on this benchmark is larger compared to FUNSD-TTA and SROIE-TTA benchmarks. Hence, we also see a greater performance improvement by using TTA/UDA across all domains and methods. For the UDA setting, DocUDA consistently outperforms adversarial-based UDA methods by a large margin, underlining the superiority of self-supervised learning and pseudo labeling in leveraging labeled and unlabeled data at test time. Also in the more challenging TTA setting, DocTTA consistently achieves the highest gain in ANLS score, with 2.57% increase on $\mathbf{E} \rightarrow \mathbf{F}$ and 17.68% on $\mathbf{F} \rightarrow \mathbf{E}$. Moreover, DocTTA significantly outperforms DANN on all domains and CDAN on 11 out of 12 adaptation scenarios, even though it does not utilize source data at test time. This demonstrates the efficacy of joint online pseudo labeling with diversity maximization and masked visual learning. Between DocUDA and DocTTA, it is expected that DocUDA performs better than DocTTA due to having extra access to source domain data. However, we observe three exceptions where DocTTA surpasses DocUDA by 1.13%, 0.79%, and 2.16% in ANLS score on $\mathbf{E} \rightarrow \mathbf{F}$ and $\mathbf{T} \rightarrow \mathbf{F}$, and $\mathbf{L} \rightarrow \mathbf{F}$, respectively. We attribute this to: i) target domain (**F**) dataset size being relatively small, and ii) large domain gap between source and target domains. The former can create an imbalanced distribution of source and target data, where the larger split (source data) dominates the learned representation. This effect is amplified due to (ii) because the two domains aren't related and the joint representation is biased in favor of the labeled source data. Another finding on this benchmark is that a source model trained on a domain with a small dataset generalizes less compared to the one with sufficiently-large dataset but has a larger domain gap with the target domain. Results for train-on-target on each domain can shed light on this. When we use the domain with the smallest dataset (**F**) as the source, each domain can only achieve its lowest ANLS score (39.70% on **E**, 24.77% on **T**, and 38.59% on **L**) whereas with **T**, second smallest domain

Table 2: ANLS scores for adapting between domains in **DocVQA-TTA** benchmark. Source-only and train-on-target serve as lower and upper bounds, respectively for performance on target domain. Standard deviations are shown in Appendix.

| Source: | Emails&Letters (**E**) | | | Figures&Diagrams (**F**) | | | Tables&Lists (**T**) | | | Layout (**L**) | | |
|---|---|---|---|---|---|---|---|---|---|---|---|---|
| Target: | **F** | **T** | **L** | **E** | **T** | **L** | **E** | **F** | **L** | **E** | **F** | **T** |
| source-only | 37.79 | 25.59 | 38.25 | 5.23 | 7.03 | 3.65 | 13.66 | 20.48 | 14.58 | 53.55 | 33.36 | 33.43 |
| DANN | 38.94 | 27.22 | 40.23 | 15.43 | 9.34 | 7.45 | 17.67 | 22.19 | 17.67 | 54.55 | 33.87 | 33.58 |
| CDAN | 39.08 | 29.33 | 41.29 | 16.99 | 11.32 | 10.23 | 27.87 | 25.23 | 27.66 | 56.82 | 34.27 | 34.81 |
| **DocUDA** (ours) | **39.23** | **43.54** | **57.99** | **24.21** | **15.76** | **20.45** | **53.19** | **29.91** | **47.81** | **61.09** | **34.85** | **41.80** |
| BN | 38.10 | 26.89 | 38.23 | 7.32 | 8.56 | 9.35 | 15.13 | 22.24 | 15.65 | 53.23 | 33.67 | 33.55 |
| TENT | 38.34 | 26.42 | 40.45 | 12.38 | 7.34 | 11.29 | 16.01 | 20.23 | 15.02 | 53.34 | 33.59 | 34.55 |
| SHOT | 38.98 | 27.55 | 39.15 | 14.34 | 10.10 | 13.21 | 22.56 | 24.33 | 19.15 | 56.23 | 34.56 | 35.65 |
| **DocTTA** (ours) | 40.36 | **35.28** | **49.35** | **22.91** | **15.67** | **16.01** | **35.67** | 30.70 | **26.32** | **59.84** | **37.01** | **39.10** |
| train-on-target | 95.28 | 93.54 | 95.01 | 39.70 | 24.77 | 38.59 | 84.59 | 70.66 | 83.73 | 92.32 | 91.36 | 93.41 |

Table 3: Ablation analysis on adapting from E to F, T, L in our DocVQA-TTA benchmark with different components including pseudo labeling, $\mathcal{L}_{MVLM}$, $\mathcal{L}_{DIV}$, and pseudo label selection mechanism using confidence only or together with uncertainty. Standard deviations are in parentheses.

| Method | F | T | L |
|---|---|---|---|
| Source-only | 37.79 (1.30) | 25.59 (1.78) | 38.25 (0.92) |
| DocTTA, conf. | 32.67 (1.68) | 21.50 (1.52) | 6.71 (3.21) |
| DocTTA, conf. & unc. | 39.45 (0.87) | 28.47 (0.72) | 47.50 (0.51) |
| DocTTA, no $\mathcal{L}_{MVLM}$ | 35.66 (0.46) | 25.72 (0.55) | 45.88 (0.34) |
| DocTTA, no $\mathcal{L}_{DIV}$ | 34.32 (0.53) | 25.17 (0.49) | 46.36 (0.21) |
| DocTTA, no pseudo labeling | 33.61 (1.65) | 23.43 (0.87) | 15.89 (1.35) |
| DocTTA | 40.36 (0.53) | 35.28 (0.76) | 49.35 (1.20) |

in dataset size in our benchmark (with 657 training documents), the scores obtained by train-on-target on **E** and **L** increases to 84.59% and 83.73%, respectively. Thus, even if we have access to entire target labeled data, the limitation of source domain dataset size is still present.

**Ablation studies:** We compare the impact of different constituents of our methods on the DocVQA-TTA benchmark, using a model trained on Emails&Letters domain and adapted to other three domains. Table 3 shows that pseudo labeling selection mechanism plays an important role and using confidence scores to accept pseudo labels results in the poorest performance, much below the source-only ANLS values and even worse than not using pseudo labeling. On the other hand, using uncertainty and raw confidence together to select pseudo labels yields the closest performance to that of the full (best) method (details are provided in Appendix). MVLM loss and diversity maximization criteria have similar impact on DocTTA's performance.

# 6   Conclusions

We introduce TTA for VDU for the first time, with our novel approach DocTTA, along with new realistic adaptation benchmarks for common VDU tasks such as entity recognition, key-value extraction, and document VQA. DocTTA starts from a pretrained model on the source domain and uses online pseudo labeling along with masked visual language modeling and diversity maximization on the unlabeled target domain. We propose an uncertainty-based online pseudo labeling mechanism that generates significantly more accurate pseudo labels in a per-batch basis. Overall, novel TTA approaches result in surpassing the state-of-the-art

TTA approaches adapted from computer vision, underlining the importance of VDU-specific TTA approach to push the the state-of-the-art in VDU.

## 7 Reproducibility

We have included the details of our experimental setup such as the utilized compute resources, pretrained model, optimizer, learning rate, batch size, and number of epochs, etc. in Sec. A.2 of the Appendix. We have provided the details of our hyper parameter tuning and our search space in Section A.2.3 of the Appendix. For our introduced benchmark datasets, the statistics of each dataset is detailed in Section A.1.2. List of the all the training and validation splits for our proposed benchmarks are also provided in `Supplemental/TTA_Benchmarks` as `json` files. To ensure full reproducibility, we will release our code upon acceptance.

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

# A Appendix

## A.1 Dataset details

### A.1.1 Licence

We use three publicly-available datasets to construct our benchmarks. These datasets can be downloaded from their original hosts under their terms and conditions:

- FUNSD Jaume et al. (2019) License, instructions to download, and term of use can be found at `https://guillaumejaume.github.io/FUNSD/work/`

- SROIE Huang et al. (2019) License, instructions to download, and term of use can be found at `https://github.com/zzzDavid/ICDAR-2019-SROIE`
- DocVQA Mathew et al. (2021) License, instructions to download, and term of use can be found at `https://www.docqa.org/datasets/doccvqa`

### A.1.2 Dataset splits

We provide the list of the documents in the *source* and *target* domains for our three benchmarks. Files are located at `Supplemental/TTA_Benchmarks/`. For FUNSD-TTA and SROIE-TTA, the validation splits have 10 and 39 documents, respectively, which are selected randomly using the seed number 42. Validation splits have a similar distribution as the source domain's training data. When performing TTA, we use the target domain data without labels – the labels are only used for evaluation purposes. Table 5 and Table 6 show the statistics of documents on source and target domains in FUNSD-TTA and SROIE-TTA, respectively.

Table 4: Number of documents in the source and target domains in FUNSD-TTA and SROIE-TTA benchmarks. We use the validation set selected from the source domain to tune TTA algorithm's hyper parameters.

Table 5: FUNSD-TTA

| | |
|---|---|
| Source Training | 139 |
| Source Validation | 10 |
| Source Evaluation, Target Training, Target Evaluation | 50 |

Table 6: SROIE-TTA

| | |
|---|---|
| Source Training | 600 |
| Source Validation | 39 |
| Source Evaluation, Target Training, Target Evaluation | 347 |

For DocVQA-TTA benchmark, we always choose 10% of source domain data for validation using the same seed (42).

Table 7: Number of documents in each domain of our DocVQA-TTA benchmark.

| | Layout (**L**) | Emails&Letters (**E**) | Tables&Lists (**T**) | Figures&Diagrams (**F**) |
|---|---|---|---|---|
| Source Training | 1807 | 1417 | 592 | 150 |
| Source Validation | 200 | 157 | 65 | 17 |
| Source Evaluation, Target Training, Target Evaluation | 512 | 137 | 187 | 49 |

### A.1.3 Text embeddings and OCR annotations

For all the benchmarks, we use officially-provided OCR annotations for each datasets. For the tokenization process, we follow Xu et al. (2020a) where they use WordPiece Wu et al. (2016) such that each token in the OCR text sequence is assigned to a certain segment of $s_i \in \{[\text{A}], [\text{B}]\}$ prepended by $[\text{CLS}]$ if it is the starting token and/or appended by $[\text{SEP}]$ if it is the ending token of the sequence. In order to have a fixed sequence length in each document, extra $[\text{PAD}]$ tokens are appended to the end, if the sequence exceeds a maximum length threshold (which is 512 in this work).

### A.2 Experiments Details

### A.2.1 Training.

We use PyTorch (Paszke et al., 2019) on Nvidia Tesla V100 GPUS for all the experiments. For **source training**, we use LayoutLMv2$_{BASE}$ pre-trained on IIT-CDIP dataset and fine-tune it with labeled source

data on our desired task. For all VDU tasks, we build task-specific classifier head layers over the text embedding of LayoutLMv2$_{BASE}$ outputs. For entity recognition and key-value extraction tasks, we use the standard cross-entropy loss and for DocVQA task, we use the binary cross-entropy loss on each token to predict whether it is the starting/ending position of the answer or not. We use AdamW (Loshchilov & Hutter, 2017) optimizer and train source model with batch sizes of 32, 32, and 64 for 200, 200, and 70 epochs with a learning rate of $5 \times 10^{-5}$ for entity recognition, key-value extraction, and DocVQA benchmarks, respectively with an exception of *Figures & Diagrams* domain on which we used a learning rate of $10^{-5}$. For BN and SHOT baselines, we followed SHOT implementation for image classification and added a fully connected layer with 768 hidden units, followed by a batch normalization layer right before the classification head. Note that we used the same backbone architecture (LayoutLMv2) as was used in DocTTA and DocUDA for all the baselines methods. For example, ResNet backbone in DANN, CDAN, and SHOT methods was replaced with LayoutLMv2 architecture. Therefore all the baselines receive the inputs exactly similar to our approach.

**Uncertainty and confidence-aware pseudo labeling** For uncertainty-aware pseudo labeling, we set a threshold ($\gamma$) above which pseudo labels are rejected to be used for training. Likewise, for confidence-aware pseudo labeling, we set a threshold for the output probability values for the predicted class *below* which pseudo labels are rejected. For the combination of the two, a pseudo label which has confidence (output probability) value above the threshold and uncertainty value (Shannon entropy) below the maximum threshold is chosen for self-training. We used confidence threshold of 0.95 and tuned the uncertainty threshold to be either 1.5 or 2 (see below).

### A.2.2 Baselines details

In this section we discuss more details about UDA and TTA baselines in our work and how they are different from our proposed approach, DocUDA and DocTTA, in both settings.

**UDA baseliens.** We compared DocUDA against **DANN** (Ganin & Lempitsky, 2015) and **CDAN** (Long et al., 2018) methods. **DANN** was proposed as a deep architecture termed as Domain-Adversarial Neural Network (DANN) for unsupervised domain adaptation which consists of a feature extractor, a label predictor, and a domain classifier. The feature extractor is similar to a generator which tries to generate domain-independed features for confusing the domain classifier. On the other hand, the domain classifier acts as the discriminator which tries to predict whether the extracted features are from the source or target domain. Lastly, the label predictor which is trained on the extracted features of the labeled source domain, tries to predict the correct class for a target sample. DANN can be trained with a special gradient reversal layer (GRL) which authors claim gives superior performance compared to when GRL it is not used in the architecture. DANN is an adversarial adaptation method and hence is substantially different from DocUDA which uses pseudo labeling and self-supervised learning in the form of masked visual language modeling. **CDAN** (Long et al., 2018) or Conditional Domain Adversarial Network uses discriminative information conveyed in the classifier predictions to assist adversarial adaptation. The key to the CDAN approach is a novel conditional domain discriminator conditioned on the cross covariance of domain-specific feature representations and classifier predictions. They further condition the domain discriminator on the uncertainty of classifier predictions, prioritizing the discriminator on easy-to-transfer examples. CDAN is another adversarial learning-based adaptation method that tries to disentangle domain specific and domain invariant features for adaptation whereas DocUDA tries to leverage the model's knowledge on both domains at the same time without disentangling or conditional learning on discriminative features.

**TTA baselines.** We compared DocTTA with **BN** (Ioffe & Szegedy, 2015; Nado et al., 2020), **TENT** (Wang et al., 2021b) and **SHOT** (Liang et al., 2020). **BN** or batch normalization is a widely used (training time) technique proposed by Ioffe & Szegedy (2015). Nado et al. (2020) adopted the training time BN and used it at test time which is a simple but surprisingly effective method and can be implemented with one line of code change. In standard practice, the batch normalization statistics are frozen to particular values after training time that are used to compute predictions. Nado et al. (2020) proposed to recalculate batch norm statistics on every batch at test time instead of re-using values from training time as a mechanism to adapt to unlabeled test data. While this is computationally inexpensive and widely adoptable to different settings, it is

not enough to overcome performance degradation issue when there is a large distribution mismatch between source and target domains. As shown in our experiments and also shown in the literature (Wang et al., 2021b;a; Chen et al., 2022) this method is usually outperformed by methods which rely on stronger adaptation mechanisms such as pseudo labeling. **TENT** (Wang et al., 2021b) proposed to adapt by test entropy minimization where the model is optimized for confidence as measured by the entropy of its predictions. TENT estimates normalization statistics and optimizes channel-wise affine transformations to update online on each batch. While this method was shown to improve BN which only updates BN statistics, it suffers from accumulation of error which can occur in the beginning of the training when predictions tend to be inaccurate especially because VDU models create a long sequence of outputs per every document resulting in a noisy training. DocTTA's training is significantly different from TENT as it primarily uses pseudo labeling, MVLM and diversity losses for adaptation. Lastly, **SHOT** Liang et al. (2020) can be considered as the closest work to ours where it combines mutual information maximization with offline clustering-based pseudo labeling. However, using simple offline pseudo-labeling can lead to noisy training and poor performance when the distribution shifts are large (Chen et al., 2022; Liu et al., 2021; Rizve et al., 2021; Mukherjee & Awadallah, 2020). In DocTTA we also use pseudo labeling but we propose online updates per batch for pseudo labels, as the model adapts to test data. Besides, we equip our method with a pseudo label rejection mechanism using uncertainty, to ensure the negative effects of predictions that are likely to be inaccurate. DocTTA also leverages masked visual langugae modeling on target domain which contributes in learning the structure of the target domain whereas SHOT only depends on pseudo labeling.

### A.2.3 Hyper parameter tuning

We use a validation set (from source domain) in each benchmark for hyper parameter tuning. Although not optimal, it is more realistic to assume no access to any labeled data in the target domain. We used a simple grid search to find the optimal set of hyper parameters with the following search space:

- Learning rate $\in \{10^{-5}, 2.5 \times 10^{-5}, 5 \times 10^{-5}\}$
- Weight decay $\in \{0, 0.01\}$
- Batch size $\in \{1, 4, 5, 8, 32, 40, 48, 64\}$
- Uncertainty threshold $\gamma \in \{1.5, 2\}$

### A.3 Measuring confidence after adaptation with DocTTA

For reliable VDU deployments, confidence calibration can be very important, as it is desired to identify when the trained model can be trusted so that when it is not confident, a human can be consulted. In this section, we focus on confidence calibration and analyze how DocTTA affects it. Figure 3 illustrates reliability diagrams for adapting from Emails & Letters (**E**) to **T**, **F**, and **L** domains in DocVQA-TTA benchmark. We compares model calibration before and after DocTTA for 'starting position index of an extracted answer' in documents. We illustrate calibration with reliability diagram (DeGroot & Fienberg, 1983; Niculescu-Mizil & Caruana, 2005), confidence histograms (Guo et al., 2017), and the ECE metric. The reliability diagram shows the expected accuracy as a function of confidence. We first group the predictions on target domain into a set of bins (we used 10). For each bin, we then compute the average confidence and accuracy and visualize them (top red plots in Fig. 3). The closer the bars are to the diagonal line, the more calibrated the model would be. Also, lower ECE values indicate better calibrations. It is observed that calibration improves with DocTTA. From this plot, we can also measure ECE as a summary metric (the lower, the better calibration). For instance, DocTTA on **E** $\rightarrow$ **L** yields significantly lower ECE, from 30.45 to 2.44. Although the reliability diagram can explain model's calibration well, it does not show the portion of samples at a given bin. Thus, we use confidence histograms (see bottom of Fig. 3) where the gap between accuracy and average confidence is indicative of calibration. Before adaptation, the model tends to be overconfident whereas, after adaptation with DocTTA, the gap becomes drastically smaller and nearly overlaps.

Table 8: Standard deviations (in parentheses) for ANLS scores shown in Table 2 of the main paper for adapting between domains in **DocVQA-TTA** benchmark (Part I).

| Source: | Emails&Letters (**E**) | | | Figures&Diagrams (**F**) | | |
|---|---|---|---|---|---|---|
| Target: | **F** | **T** | **L** | **E** | **T** | **L** |
| Source-only | 37.79 (1.30) | 25.59 (1.78) | 38.25 (0.92) | 5.23 (2.86) | 7.03 (2.42) | 3.65 (2.76) |
| DANN | 38.94 (1.20) | 27.22 (1.32) | 40.23 (1.78) | 15.43 (1.34) | 9.34 (3.23) | 7.45 (3.29) |
| CDAN | 39.08 (1.59) | 29.33 (0.82) | 41.29 (2.80) | 16.99 (1.56) | 11.32 (2.43) | 10.23 (2.54) |
| **DocUDA** (ours) | **39.23** (1.42) | **43.54** (0.91) | **57.99** (0.12) | **24.21** (0.72) | **15.76** (0.67) | **20.45** (0.43) |
| BN | 38.10 (1.01) | 26.89 (0.59) | 38.23 (0.98) | 7.32 (2.43) | 8.56 (2.34) | 9.35 (3.21) |
| TENT | 38.34 (0.74) | 26.42 (0.52) | 40.45 (0.81) | 12.38 (3.12) | 7.34 (2.54) | 11.29 (2.45) |
| SHOT | 38.98 (0.89) | 27.55 (0.81) | 39.15 (1.23) | 14.34 (3.87) | 10.10 (1.34) | 13.21 (2.54) |
| **DocTTA** (ours) | **40.36** (0.53) | **35.28** (0.76) | **49.35** (1.20) | **22.91** (0.45) | **15.67** (0.78) | **16.01** (1.18) |
| Train-on-target | 95.28 (1.32) | 93.54 (0.91) | 95.01 (1.34) | 39.70 (1.02) | 24.77 (0.23) | 38.59 (0.78) |

Table 9: Standard deviations (in parentheses) for ANLS scores shown in Table 2 of the main paper for adapting between domains in **DocVQA-TTA** benchmark (Part II).

| Source: | Tables&Lists (**T**) | | | Layout (**L**) | | |
|---|---|---|---|---|---|---|
| Target: | **E** | **F** | **L** | **E** | **F** | **T** |
| Source-only | 13.66 (1.67) | 20.48 (1.54) | 14.58 (1.30) | 53.55 (1.76) | 33.36 (1.84) | 33.43 (1.94) |
| DANN | 17.67 (1.49) | 22.19 (2.34) | 17.67 (2.58) | 54.55 (0.72) | 33.87 (1.02) | 33.58 (0.28) |
| CDAN | 27.87 (1.82) | 25.23 (1.24) | 27.66 (2.62) | 56.82 (0.62) | 34.27 (0.82) | 34.81 (0.43) |
| **DocUDA** (ours) | **53.19** (0.92) | **29.91** (0.45) | **47.81** (0.41) | **61.09** (0.08) | **34.85** (0.16) | **41.80** (0.06) |
| BN | 15.13 (2.04) | 22.24 (1.29) | 15.65 (2.43) | 53.23 (1.08) | 33.67 (1.17) | 33.55 (1.55) |
| TENT | 16.01 (2.83) | 20.23 (2.61) | 15.02 (2.83) | 53.34 (0.93) | 33.59 (1.82) | 34.55 (1.44) |
| SHOT | 22.56 (1.12) | 24.33 (2.54) | 19.15 (2.39) | 56.23 (1.20) | 34.56 (1.02) | 35.65 (1.23) |
| **DocTTA** (ours) | **35.67** (1.10) | **30.70** (0.80) | **26.32** (1.27) | **59.84** (0.04) | **37.01** (0.05) | **39.10** (0.06) |
| Train-on-target | 84.59 (1.02) | 70.66 (0.82) | 83.73 (0.23) | 92.32 (0.01) | 91.36 (0.04) | 93.41 (0.05) |

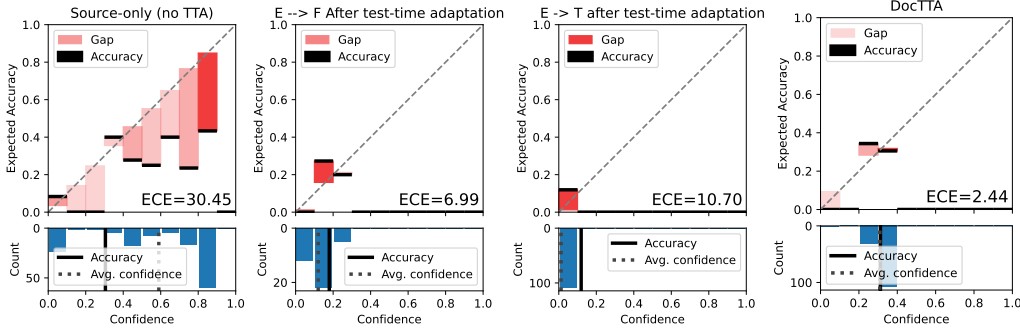

Figure 3: Comparing confidence calibration with and without DocTTA when adapting from Emails & Letters domain to other domains in DocVQA-TTA benchmark.

## A.4 Standard deviations

Here we show the standard deviations obtained over 3 seeds for results reported in Table 2 of the main paper in Table 8 (part I) and 9 (part II), respectively.

### A.5 DocUDA algorithm

In DocUDA, we use source data during training on target at test time. Therefore, DocUDA has an additional objective function which is a cross-entropy loss using labeled source data:

$$\mathcal{L}_{CE_{Src}}(\theta_t) = -\mathbb{E}_{x_s \in \mathcal{X}_s} \sum_{c=1}^{C} y_c \log \sigma(f_t(x_s)), \tag{6}$$

And the overall objective function of DocUDA is Eq. 5 plus Eq. 6:

$$\mathcal{L}_{\text{DocUDA}} = \mathcal{L}_{MVLM} + \mathcal{L}_{CE} + \mathcal{L}_{DIV} + \mathcal{L}_{CE_{Src}}. \tag{7}$$

Algorithm 2 shows how DocUDA is performed on VDU tasks.

---

**Algorithm 2** DocUDA for closed-set UDA in VDU

---

1: **Input:** labeled source documents $\{x_s^{(i)}, y_s^{(i)}\}_{i=1}^{n_s}$, target documents $\{x_t^i\}_{i=1}^{n_t}$, test-time training epochs $n_e$, test-time training learning rate $\alpha$, uncertainty threshold $\gamma$
2: **Initialization:** Initialize target model $f_{\theta_t}$ with LayoutLMv2$_{BASE}$ weights trained on IIT-CDIP dataset.

3: **for** $epoch = 1$ to $n_e$ **do**
4:     Perform masked visual-language modeling in Eq. 1
5:     Generate pseudo labels and accept a subset using criteria in Eq. 2 and fine-tune with Eq. 3
6:     Maximize diversity in pseudo label predictions Eq. 4
7:     Perform supervised training using labeled source data with Eq. 3
8:     $\theta_t \leftarrow \theta_t - \alpha \nabla \mathcal{L}_{\text{DocUDA}}$          ▷ Update $\theta_t$ via total loss in Eq. 7
9: **end for**

---

### A.6 Qualitative Results

Here we show a randomly selected document from our FUNSD-TTA benchmark. Comparing the results before and after using DocTTA shows that our method can refine some wrong predictions made by the unadapted model.

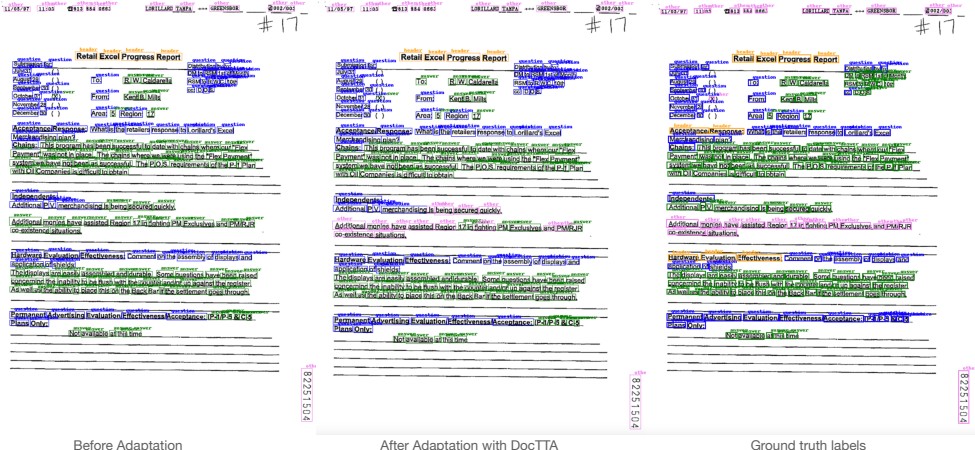

Before Adaptation        After Adaptation with DocTTA        Ground truth labels

Figure 4: From left to right we show predictions made by (i) an unadapted model, (ii) after using DocTTA, iii) ground truth labels.

### A.7 Additional Baseline Results

Here we show the performance of our model against AdaContrast (Chen et al., 2022) for TTA and SHOT-UDA (Liang et al., 2020) on DocVQA-TTA benchmark when adapting from *Emails & Letters* domain to **F**, **T**, and **L**.

Table 10

| Source | Emails&Letters (E) | | |
|---|---|---|---|
| Target | **F** | **T** | **L** |
| SHOT (UDA) | 39.02 | 31.35 | 48.87 |
| **DocUDA** (ours) | **39.23** | **43.54** | **57.99** |
| AdaContrast (TTA) | 37.21 | 27.43 | 38.69 |
| **DocTTA** (ours) | **40.36** | **35.28** | **49.35** |

### A.8 Layout Illustration

In our method, layout refers to a bounding box associated with each word in the text input sequence and is represented with a 6-dimensional vector in the form of $(x_{min}, x_{max}, y_{min}, y_{max}, w, h)$. where $(x_{min}, y_{min})$ corresponds to the position of the lower left corner and $(x_{max}, y_{max})$ represents the position of the upper right corner of the bounding box and $w$ and $h$ denote the width and height of the box, respectively as shown below

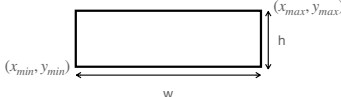

Figure 5: Illustration of layout as a bounding box associated with each word in the text input sequence.

