# OpenReview forum: "Test-Time Adaptation for Visual Document Understanding"
_TMLR — Accepted by TMLR_

### Review · Reviewer_JqyT · 2023-04-08

**Summary Of Contributions:**

The paper aims to do domain adaptation (adapting models trained on a source main to a target domain) using unlabeled target-domain document data. The approach assumes that optical character recognition is already performed, so we already have information on text and bounding boxes. Then, the approach involves
- (1) masked visual language modeling,
- (2) pseudo labeling: specifically, first performing per-batch pseudo-labeling and then performing uncertainty-aware (based on thresholds according to Shannon entropy) selection according to Equations 2-3,
- (3) maximizing diversity of pseudo-labels in case the pseudo-labels get dominated by one or very few classes only.

The tasks involve entity recognition, key-value extraction, and document visual question answering. The tasks are created by the authors using existing datasets: specifically, they split existing datasets to a source domain and a target domain.
- For the entity-recognition adaptation task (“FUNSD-TTA”), an existing dataset is split into the source domain containing forms with more texts and the target domain containing forms that are only sparsely filled.
- For the key-value extraction adaptation task (“SROIE-TTA”), the source domain contains standard-looking receipts (more description in the paper), and the target domain contains blurry or rotated or colored or other non-standard-looking receipts.
- For document VQA adaptation (“DocVQA-TTA”), the dataset is split into different categories according to topic/domain, and then combined into source vs. target domains.

Results are better than state-of-the-art domain adaptation approaches according to automatic metrics: the first two tasks use entity-level F1, and the third task uses average normalized Levenshtein.


**Audience:**

Yes

**Claims And Evidence:**

Yes

**Requested Changes:**

Addressing the above concerns.

Overall I think the work is quite sound. I'm also curious: what are the other potential impactful use cases of the authors' approach (other than the three described tasks)?

**Strengths And Weaknesses:**

Strengths
- Interesting topic. Domain adaptation in visual document understanding has real frequent use cases in the real world.
- The tasks build on prior datasets, and the source vs. target domain split is clever.
- Uncertainty-based selection mechanism (the second part of the approach) seems very crucial.
- It’s interesting that the ablation which removes the third objective (maximizing the diversity of pseudo-labels) has much poorer results.
- It’s quite surprising to me that the ablation which removes masked visual LM training is better than the ablation that only removes pseudo labeling.
- Writing is clear.

Concerns

Question: No coefficient for Equation (5) when combining different objectives. Would the authors expect that different coefficients would make the results better/worse?

Section 3.2, objective II: “We empirically observe that raw confidence values are overconfident despite being right or wrong.” Is there evidence for this statement? Apologies I’m missing the details.

Algorithm 1: The equations don’t seem to correspond to the description. Should Equation 2 and Equation 3 be swapped?

The tasks have synthetically and manually designed splits, and it’s unclear whether a method that performs well on the proposed three tasks would imply that the method performs well on the noisier real-world test-time domain shifts. But this is not a huge concern imo.

---

> ### Author Response · Authors · 2023-04-15
> **Responses to Reviewer JqyT [1/2]**
>
> We thank the reviewer for their positive feedback and their precise summary of our contributions and appreciating our method and datasets, particularly acknowledging that our paper addresses an “interesting topic” which “has real frequent use cases in the real world” and “the source and target domain splits in the benchmark datasets is clever” and that our “writing is clear”.
>
> We provide answers to reviewer’s concerns and requested changes below:
>
> `Question: No coefficient for Equation (5) when combining different objectives. Would the authors expect that different coefficients would make the results better/worse?`
>
> This is a great point! First, we note that unsupervised hyper parameter tuning is an open research problem as approximating performance without relying on any labeled data is fundamentally very challenging. Improvements in unsupervised hyper parameter tuning, especially on how to pick many hyper parameters jointly, can potentially bring benefits to test-time adaptation (or any other setting in which it is not realistic to assume any labeled data is available, such as unsupervised anomaly detection) [1].
>
> In test-time adaptation, we assume that there is no labeled data available from the target domain. Representing a more practical scenario, as we mentioned in Sec A.2.2 of the Appendix (Hyper parameter tuning section), we assume that a small amount of data from the source domain can be split as validation data to be used for hyperparameter tuning, without access to any labeled data in the target domain. We note that this way of doing hyperparameter tuning is not as effective as standard supervised learning setting so we propose that it should be applied in a more constrained way and should merely focus on the most key parameters, and thus, we employ a very simple grid search to find learning rate, weight decay, batch size, and uncertainty threshold as our hyper parameters.
>
> One can add the loss terms’ coefficients to the given hyperparameter search space as well. We believe finding optimal values for them can bring performance gains, however, the gains might not be significant and in some cases might even be negative, as optimizing loss values on the source domain might not help with generalization to the target domain. Therefore, since all the losses on the target domain were of the same order of magnitude, we kept the coefficients as 1 to not overfit to the source domain.
>
> [1] Saito, Kuniaki, et al. "Tune it the right way: Unsupervised validation of domain adaptation via soft neighborhood density." Proceedings of the IEEE/CVF International Conference on Computer Vision. 2021.
>
> `Section 3.2, objective II: “We empirically observe that raw confidence values are overconfident despite being right or wrong.” Is there evidence for this statement? Apologies I’m missing the details.`
>
> Yes, indeed there are several papers that have explored the issues related to neural networks being overconfident, starting with Guo et al. [2] followed up by others [3-6]. We also observed this in our experiments especially with larger distribution shifts between the source and target domain. We have added these references to our statement in the paper.
>
> [2] Guo, Chuan, et al. "On calibration of modern neural networks." International conference on machine learning. PMLR, 2017.
>
> [3] Wei, Hongxin, et al. "Mitigating neural network overconfidence with logit normalization." International Conference on Machine Learning. PMLR, 2022.
>
> [4] Meinke, Alexander, and Matthias Hein. "Towards neural networks that provably know when they don't know." arXiv preprint arXiv:1909.12180 (2019).
>
> [5] Thulasidasan, Sunil, et al. "On mixup training: Improved calibration and predictive uncertainty for deep neural networks." Advances in Neural Information Processing Systems 32 (2019).
>
> [6] Kristiadi, Agustinus, Matthias Hein, and Philipp Hennig. "Being bayesian, even just a bit, fixes overconfidence in relu networks." International conference on machine learning. PMLR, 2020.
>
>
> `Algorithm 1: The equations don’t seem to correspond to the description. Should Equation 2 and Equation 3 be swapped?`
>
> Thank you for bringing our attention to this typo in Algorithm 1. We have corrected Line 5 in the uploaded version to “Generate pseudo labels and accept a subset using criteria in Eq. 2 and fine-tune with Eq. 3”

---

> > ### Author Response · Authors · 2023-04-15
> > **Responses to Reviewer JqyT [2/2]**
> >
> > `The tasks have synthetically and manually designed splits, and it’s unclear whether a method that performs well on the proposed three tasks would imply that the method performs well on the noisier real-world test-time domain shifts. But this is not a huge concern imo.`
> >
> > We agree with the reviewer that the real-world dataset shifts for document data can be very diverse. Our primary motivation with the proposed benchmarks is to propose a diverse evaluation suite that can reflect real-world performance of the adaptation methods - that’s why we used different datasets and different ways of splitting criteria. The proposed shifts are also inspired by what we have empirically found in our experiences from real-world document data that causes performance degradation between training and test time.
> >
> > `I'm also curious: what are the other potential impactful use cases of the authors' approach (other than the three described tasks)?`
> >
> > This is a great question and we hope followup works will focus on them and make more impact in this domain. Some candidates we have in mind are:
> > - document/ID verification for new types of documents (e.g. a bank is expanding to a new country/state),
> > - newly introduced formats for the types of documents with a lot of labeled data (e.g. new versions of tax forms),
> > - better addressing the quality gaps coming from visual sensor differences (e.g. a new machine type for scanning) as the deployed hardware versions evolve faster.

---

> > > ### Comment · Reviewer_JqyT · 2023-05-01
> > > **Response to authors**
> > >
> > > Thanks for the response.
> > >
> > > I read through other reviewers' feedback as well as the discussion. I am not concerned about the work's novelty -- test-time adaptation is not often studied but has broad real-world use cases. Piecing together different strategies like masked visual/language modeling and uncertainty-based approaches takes effort, and experiments and results are convincing. It's good to see that the authors' approach outperforms SHOT (a much more recent baseline) as well.
> > >
> > > Although I was slightly concerned about the fact that the designed tasks do not necessarily exactly correspond to real-world domain shifts, I think the tasks are still meaningful (and quite different from previous benchmarks that do not address test-time domain shifts catering toward real-world applications).

---

### Review · Reviewer_2HCZ · 2023-04-08

**Summary Of Contributions:**

This paper proposes DocTTA, a test-time adaptation algorithm for visual document understanding (VDU) which effectively adapts representations learned from self-supervised learning algorithms to source-free domain adaptation settings in new target domains. The authors propose new benchmark datasets for this task and demonstrate the efficacy of their approach.

**Audience:**

Yes

**Claims And Evidence:**

Yes

**Requested Changes:**

- When looking at Table 2, the discrepancies in ANLS scores between all the existing approaches and train-on-target are quite large. Would the authors address this in the text?
- Elaborating a bit more on why they decided to adapt methods such as DANN and CDAN as their baselines would be helpful.
- Talking a bit more in detail and clarifying the pseudo-labeling component of the adaptation procedure would be helpful. Section 3.2 was a bit difficult to parse.

**Strengths And Weaknesses:**

Strengths:
- The problem is important and relatively under-explored in the field. I can see this work sparking more interest in this domain.
- Most of the paper is clearly written and easy to read. For example, the explanation for how the pre-training happens (via the masked visual language modeling task, self-training with pseudolabels, and diversity objective), were straightforward.
- The authors also introduce new benchmarks for the VDU domain adaptation task that are constructed from existing, publicly available data - FUNSD for entity recognition, SROIE for key-value extraction, and DocVQA for visual question-answer.
 - The authors provided extensive detail on how the datasets were constructed, their design decisions on how to construct the datasets, etc.

Weaknesses:
- Although there do not seem to exist relevant baselines to compare DocTTA against, I'm curious why the authors decided to adapt methods from quite a while back as their baselines (e.g. DANN, which is from 2015).
- The authors mention in the paper (as well as in their ablation studies) that the way in which they handled the pseudo-labeling is quite important for achieving good performance, but it's not clear to me why this is the case.

---

> ### Author Response · Authors · 2023-04-15
> **Response to Reviewer 2HCZ [1/2]**
>
> We thank the reviewer for their positive feedback and appreciating our method and datasets, particularly acknowledging that our paper addresses an “important and relatively under-explored problem” and that our work “sparks more interest in this domain”, “is mostly clearly written and easy to read” and that we “provided extensive detail on how the datasets were constructed and on our design decisions”.
>
> Here, we address the weakness points and requested changes by the reviewer:
>
> `The gap between ANLS scores for “ train-on-target” and all other methods is quite large.`
>
> We note that the “train-on-target” baseline does not adhere to any domain adaptation setting, but it merely serves as an upper bound for performance obtained in the hypothetical scenario of training and testing with the ground truth test labels on the target domain. On the other hand, for DocTTA and all other baselines, we consider the actual real-world scenario that they don’t have access to test labels. Thus, the observed large gap is expected because of the setup difference. This shows the severity of the distribution shifts, and also the room for improvement if we have methods that can take advantage of the labeled target data, if they become available. Details about this baseline are given in Sec 5 (Baselines). We have further clarified it by adding a note in the captions of Table 1 and 2.
>
>
> `Elaborating a bit more on why they decided to adapt methods such as DANN and CDAN as their baselines would be helpful.`
>
> Our emphasis in this paper is on the harder task of test-time adaptation setting where we do not have access to source domain data during adaptation. On the other hand, domain adaptation settings consider the scenario where source data are jointly optimized with unlabeled target data. We consider DANN and CDAN as the UDA baselines as they are commonly used in practice and are well established. We believe more advanced UDA methods can further improve their results. We have added a note about this.
>
> We have also shown results with a more recent UDA method (SHOT-UDA) in the Appendix Sec A.7 where we have compared DocUDA with a UDA version of SHOT method from ICML 2020. DocTTA even outperforms SHOT baseline as well.
>
> Overall, the takeaway we would like to convey is that a judiciously-designed method for test-time adaptation setting can outperform commonly-used methods in domain adaptation setting. The hypothetically-optimal domain adaptation method would have the potential to outperform the optimal test-time adaptation method as it fundamentally utilizes more information that comes from the source data, but this benefit is more clear if the source and target data are from similar distributions.
>
>
> `Talking a bit more in detail and clarifying the pseudo-labeling component of the adaptation procedure would be helpful. Section 3.2 was a bit difficult to parse.`
>
> We have improved the writing in Sec 3.2 (Objective II). In summary, this section has three main takeaways:
>
> i) We use an “online” pseudo labeling mechanism where we generate pseudo labels per batch while also optimizing the MVLM and diversity losses in the same batch.
>
> ii) We filter pseudo labels based on Shannon's entropy value as a measure of uncertainty, i.e., we only train with “certain” pseudo labels whose uncertainty is below a specific threshold.
>
> iii) The difference between our pseudo labeling mechanism and prior works which used pseudo labeling for test-time adaptation in image classification tasks is as follows:
> - We use “per-batch” pseudo labels to use the latest version of the model as opposed to the works that generate pseudo labels “per-epoch” (Liang et al. 2020; 2021 Wang et al., 2021a)
> - We directly use model’s prediction to generate pseudo labels as opposed to the works that use a clustering mechanism (Liang et al. 2020; 2021 Wang et al., 2021a)
> - We filter pseudo labels based on their uncertainty whereas in (Liang et al. 2020; 2021 Wang et al., 2021a) no filtering mechanism is deployed. (Rizve et al. 2021) uses pseudo labeling as a semi-supervised learning method (not for adaptation) where they filter pseudo labels based on raw confidence values (softmax probabilities) and uncertainty measurements using MC-Dropout.

---

> > ### Author Response · Authors · 2023-04-15
> > **Response to Reviewer 2HCZ [2/2]**
> >
> > `The authors mention in the paper (as well as in their ablation studies) that the way in which they handled the pseudo-labeling is quite important for achieving good performance, but it's not clear to me why this is the case.`
> >
> > It is important to filter the generated pseudo labels (especially under distribution shifts) because noisy labels can hurt the performance, as the model can overfit the information from them. To provide deep insights into the impact of pseudo-labeling mechanism for DocTTA, we compare three different filtering mechanisms for pseudo labels in ablation studies: i) confidence ii) confidence and uncertainty, and iii) uncertainty, with confidence being based on the Softmax score and uncertainty being measured with Shannon’s entropy. Our results show that filtering based on confidence alone yields the poorest results (even worse than a model that is not adapted (Source-only)) while using confidence together with uncertainty is suboptimal, compared to the best results achieved by using uncertainty only. Overall, the judicious design of pseudo-labeling along with an appropriate filtering mechanism, is very important to achieve superior test-time adaptation results.

---

### Review · Reviewer_hWWC · 2023-04-09

**Summary Of Contributions:**

- This work introduces a new benchmark for domain adaptation for  document understanding tasks such as form understanding and VQA.
- As the authors observe domain adaptation for document images understanding is not a well studied problem. Authors show that a simple in-domain self supervised training using MLVM  and retraining using pseudo labels can help to some extent.
-


**Audience:**

Yes

**Broader Impact Concerns:**

The paper does not include any statement(s) on the ethical implications. This reviewer could not think of any.

**Claims And Evidence:**

Yes

**Requested Changes:**

- In page 2 authors write "pseudo-labelling can result in accumulation of errors.." in the case of VDU. Can the authors please explain this in the context of VQA on document images. This reviewer could not understand what is written above it either.
- Authors use a baseline from Nado et al. that uses batch normalization. But this work is not discussed anywhere in the related works. Request the authors to add details of of all the three baselines and how it compares with the approach proposed in this work.
- In Fig2, align the text on the left with the  rows that show different embedding. For example, 1D pos. emb. is shown against the segment embedding

-  It will be helpful if an illustration of DocTTA framework for DocVQA  is provided as it comes with an extra input in the form of a question. The current Figure only shows an image in the input.
- In page 5, it is written that layout X^B is a 6 dimensional vector. When question tokens are introduced, what will be this vector be, for a question token?
- One aspect that this reviewer is confused about is how the training happens for DocTTA. Is  MVLM and supervised training using MVLM happening concurrently ?  If that is the case, this reviewer has following questions in this regard

1. For a token classification style task on datasets like FUNSD, if a token from the document is maksed, how can the model generate a label for it  (label as in the label corresponding to the semantic entity)

2. For the VQA task, when some one tokens are masked does the model have enough information to find the answer to the question

3. Would not it be better to perform the MVLM first, and then use pseudo-label based supervised training. Will it be possible for the authors to perform this experiment ?

- In Algorithm 1, Input takes in only target documents. However for VQA, would not question also be a part of the input?
- Page 6. in section 3.3, third sentence, there is a typo.
-  How are the baselines used in this work modified/adapted for the problem of VQA. This reviewer feels that more details on what are these baselines and how are they used for the VDU problem needs to be added


**Strengths And Weaknesses:**

### Strengths

- The empirical results show that the proposed , unsupervised approach  that doesn't use source data helps to improve the performance on the target domain.
- Paper is generally well written

### Weaknesses
- The proposed approach is a trivial application of existing methods for domain adaptation in document image understanding. The proposed approach is essentially an in-domain self supervised  training using MVLM  on the target data and a supervised training using pseudo labels. Although these experiments are worthy contributions,  the technical contribution is limited.  MVLM based pretraining is quite popular in vision and language community. Self supervised learning using this objective, on the target domain is a a trivial contribution. The second objective is training using pseudo labels, which is widely used in domain adaptation literature.
- For the FUNSD-TTA, what is the rationale behind the splitting criteria - sparse vs dense text. How does that make a "shift" from domain adaptation perspective ? Similarly for SROIE-TTA, the criteria for the split is more or less intuitive and not backed by any supporting evidence or reasoning.
-

---

> ### Author Response · Authors · 2023-04-15
> **Responses to Reviewer hWWC [1/3]**
>
> We thank the reviewer for their positive comments and useful feedback that has helped us to improve our paper.
>
> We provide answers to “weaknesses” and “requested changes” below:
>
> `The proposed approach is a trivial application of existing methods for domain adaptation in document image understanding. The proposed approach is essentially an in-domain self supervised training using MVLM on the target data and a supervised training using pseudo labels. Although these experiments are worthy contributions, the technical contribution is limited. MVLM based pretraining is quite popular in vision and language community. Self supervised learning using this objective, on the target domain is a a trivial contribution. The second objective is training using pseudo labels, which is widely used in domain adaptation literature.`
>
> First, we would like to underline that for visual document understanding, we focus on the problem of test-time adaptation (TTA), which is different from conventional (unsupervised) domain adaptation settings. Domain adaptation considers joint optimization using data from source and target domains, whereas for TTA, source data are not included in the training at test time.
>
> Regarding how DocTTA distinguishes from other TTA methods, there are two major aspects:
> DocTTA is the only TTA approach that combines multimodal self-supervised representation learning with pseudo labeling. Other pseudo labeling based TTA approaches proposed for image classification task, either employing pseudo labeling alone (eg. SHOT) or combining contrastive learning as the self-supervision with pseudo labeling (eg. AdaContrast by Chen et al, CVPR 22).
> DocTTA also introduces a new way to select and filter pseudo labels for the purpose of TTA for documents. Unlike the SHOT method that does not filter pseudo labels, DocTTA proposes using Shannon’t entropy as a measure of uncertainty to select certain pseudo labels only. This is also different from (Rizve et al. 2021) which focuses on performing semi-supervised learning at computer vision tasks (image and video classification) with pseudo labeling so that they can bridge the performance gap between consistency regularization methods and pseudo labeling based methods. For that, they show a pseudo-labeling selection mechanism that relies on both confidence (softmax output probabilities) and uncertainty (measured by MC-Dropout). In our paper, the focus is on TTA in multimodal setting (not just for image data but language and layout information), for which we use pseudo labeling (along with MVLM and diversity loss) and we show that the most effective pseudo labeling selection mechanism is the one that relies on uncertainty measured with Shannon’s entropy only.
>
> `For the FUNSD-TTA, what is the rationale behind the splitting criteria - sparse vs dense text. How does that make a "shift" from domain adaptation perspective ? Similarly for SROIE-TTA, the criteria for the split is more or less intuitive and not backed by any supporting evidence or reasoning.`
> Our primary motivation with the proposed benchmarks is to propose a diverse evaluation suite that can reflect real-world performance of the adaptation methods - that’s why we used different datasets and different ways of splitting criteria. The proposed shifts are inspired by what we have empirically found in our experiences from real-world document data that causes performance degradation between training and evaluation scenarios. The reason we used FUNSD (Form Understanding in Noisy Scanned Documents) for sparsity is that this dataset is created with noisy data (e.g. originating from the disparity in scanning processes). So the shift between the source and target is the sparsity of available information in the documents. Our motivation for SROIE dataset was visual appearance discrepancy (ink color, additional hand written text, etc) and different content and/or layout in the receipts.

---

> > ### Author Response · Authors · 2023-04-15
> > **Responses to Reviewer hWWC [2/3]**
> >
> > `In page 2 authors write "pseudo-labelling can result in accumulation of errors.." in the case of VDU. Can the authors please explain this in the context of VQA on document images. This reviewer could not understand what is written above it either.`
> >
> > Pseudo labeling is the process of using a model to make predictions, and using those predictions and treating them as “labels” to train the model further. When a model is used to generate pseudo labels on a distribution which is different from what it was trained on, it is more prone to make wrong predictions due to the distribution gap between the source and target domains. “Naive pseudo labeling can result in accumulation of errors” is referring to generating noisy pseudo labels by such a model, which would result in training with wrong labels and thus model accumulating its mistakes.
> >
> > While this can be applied to all VDU tasks, we make an example of this in the VQA task as follows. Consider a model trained on answering questions in Emails domain (source) to adapt to Figures domain (target) where it is supposed to find answers in given figures while it has only seen contents of emails. This model is most likely prone to making lots of mistakes in finding the correct starting/ending positions of answers regarding figures and if we do not filter out those noisy pseudo labels, the model can never make any progress in the target domain.
> >
> > `Authors use a baseline from Nado et al. that uses batch normalization. But this work is not discussed anywhere in the related works. Request the authors to add details of of all the three baselines and how it compares with the approach proposed in this work.`
> >
> > Thank you for this note. We have added the reference for the method that uses batch normalization (BN) to the related work section. We have also added extensive details for all the baselines in the Appendix (Sec. A.2.2).
> >
> > `In Fig2, align the text on the left with the rows that show different embedding. For example, 1D pos. emb. is shown against the segment embedding`
> > Thank you for pointing this out. We have aligned the text boxes in Fig 2 in the revised PDF.
> >
> > `It will be helpful if an illustration of DocTTA framework for DocVQA is provided as it comes with an extra input in the form of a question. The current Figure only shows an image in the input.`
> > We have updated Figure 2 with illustrating the input data structures required for all the three VDU tasks we have considered in the paper.
> >
> > `In page 5, it is written that layout X^B is a 6 dimensional vector. When question tokens are introduced, what will be this vector be, for a question token?`
> >
> > X^B is a vector representing a single bounding box layout (as shown in Appendix A.8) with (xmin, xmax, ymin, ymax, w, h). For DocVQA task the input is composed of the question, words, and bounding boxes which are all concatenated together and get tokenized using pretrained LayoutLMv2 tokenizer. We have added a note about this in Sec 3.1 where we explain the inputs of the model.
> >
> > `One aspect that this reviewer is confused about is how the training happens for DocTTA. Is MVLM and supervised training using MVLM happening concurrently ? If that is the case, this reviewer has following questions in this regard`
> >
> > As discussed in page 6 (Objective II) also shown in Eq 5 and in the Algorithm 1, all losses are added together to form a single DocTTA objective, that is used for adaptation for each batch. We also note that in our setting, there is no labeled data from the target domain, and hence there is no supervised training (which uses labels).
> >
> > `For a token classification style task on datasets like FUNSD, if a token from the document is maksed, how can the model generate a label for it (label as in the label corresponding to the semantic entity)`
> >
> > If it is masked then it is not included in the cross-entropy loss calculation because the attention mask for it is zero so the model essentially cannot make a prediction for it in that particular batch. We have added a note about this in the paper below Eq. 3.
> >
> >
> > `For the VQA task, when some one tokens are masked does the model have enough information to find the answer to the question`
> > Yes, we know the starting and the ending index of the answer in each document and we use it to not mask words in between them. Also, we only mask 15% of the tokens (similar to the original BERT model) and trying to extract information from partially available information enhances the model’s ability to predict the words on the target domain which helps with finding the answer in the document.

---

> > > ### Author Response · Authors · 2023-04-15
> > > **Responses to Reviewer hWWC [3/3]**
> > >
> > > `Would not it be better to perform the MVLM first, and then use pseudo-label based supervised training. Will it be possible for the authors to perform this experiment ?`
> > >
> > > This is a great question indeed. We also experimented with employing MVLM first before pseudo labeling where we collected pseudo labels before training MVLM as training MVLM separately overrides the weights learned on the source domain, resulting in forgetting issue such that the majority of pseudo labels generated right after MVLM training would be wrong. Below, we show the results for training MVLM first followed by pseudo labeling after. It can be seen that performance degrades substantially compared to the proposed version of DocTTA with all objectives being trained together at the same time.
> > > | Source                                                                   |       |       | Emails&Letters (E) |
> > > |--------------------------------------------------------------------------|:-----:|:-----:|:------------------:|
> > > | Target                                                                   | F     | T     | L                  |
> > > | DocTTA w/ MVLM followed by Pseudo labeling and diversity loss            | 38.12 | 31.24 | 43.29              |
> > > | DocTTA w/ optimizing all loss terms together (as presented in the paper) | 40.36 | 35.28 | 49.35              |
> > >
> > >
> > > `In Algorithm 1, Input takes in only target documents. However for VQA, would not question also be a part of the input?`
> > >
> > > Yes, in the VQA task, questions are part of the input. We have depicted this now in the updated version of Fig 2 and we have included questions in the inputs of Algorithm 1.
> > >
> > > `Page 6. in section 3.3, third sentence, there is a typo.`
> > > Thank you for pointing that out. We have removed the word “provides” and kept “can provide”.
> > >
> > > `How are the baselines used in this work modified/adapted for the problem of VQA. This reviewer feels that more details on what are these baselines and how are they used for the VDU problem needs to be added`
> > >
> > > VQA task is different in its inputs and outputs, which are both independent of the adaptation mechanism so essentially there is no methodological difference when it comes to application of the proposed DocTTA. Inputs include questions as part of the input sequence and outputs are the starting and the ending position IDs of the answer in the document.
> > >
> > > Regarding the baselines, they were all introduced for image classification tasks with cross-entropy loss, and hence we had to change them to use a binary cross-entropy loss on each token to predict whether it is the starting/ending position of the answer or not. We have clarified this further in Appendix Sec. A.2.1 and A.2.2.

---

> > ### Comment · Reviewer_hWWC · 2023-04-17
> > **hWWC's response to authors' response**
> >
> > ```
> > Regarding how DocTTA distinguishes from other TTA methods, there are two major aspects: DocTTA is the only TTA approach that combines multimodal self-supervised representation learning with pseudo labeling. Other pseudo labeling based TTA approaches proposed for image classification task, either employing pseudo labeling alone (eg. SHOT) or combining contrastive learning as the self-supervision with pseudo labeling (eg. AdaContrast by Chen et al, CVPR 22). DocTTA also introduces a new way to select and filter pseudo labels for the purpose of TTA for documents. Unlike the SHOT method that does not filter pseudo labels, DocTTA proposes using Shannon’t entropy as a measure of uncertainty to select certain pseudo labels only. This is also different from (Rizve et al. 2021) which focuses on performing semi-supervised learning at computer vision tasks (image and video classification) with pseudo labeling so that they can bridge the performance gap between consistency regularization methods and pseudo labeling based methods. For that, they show a pseudo-labeling selection mechanism that relies on both confidence (softmax output probabilities) and uncertainty (measured by MC-Dropout). In our paper, the focus is on TTA in multimodal setting (not just for image data but language and layout information), for which we use pseudo labeling (along with MVLM and diversity loss) and we show that the most effective pseudo labeling selection mechanism is the one that relies on uncertainty measured with Shannon’s entropy only.
> > ```
> > What authors highlight as differences compared to existing TTA are i) it is the only approach that combines multimodal self-supervised representation learning, However this reviewer would like to reiterate that this is nothing but MVLM that is been widely used in BERT-like models like LayoutLM, VLBERT, VILBERT etc. Secondly, using entropy to pick pseudo labels is a popular technique. More the confusion,  higher the change of error is a general rule applied to pick pseudo  labels.  For example see paper https://openreview.net/pdf/c979bcaed90f2b14dbf27b5e90fdbb74407f161b.pdf , section 3.3.
> >
> > ```
> > Our primary motivation with the proposed benchmarks is to propose a diverse evaluation suite that can reflect real-world performance of the adaptation methods - that’s why we used different datasets and different ways of splitting criteria. The proposed shifts are inspired by what we have empirically found in our experiences from real-world document data that causes performance degradation between training and evaluation scenarios. The reason we used FUNSD (Form Understanding in Noisy Scanned Documents) for sparsity is that this dataset is created with noisy data (e.g. originating from the disparity in scanning processes). So the shift between the source and target is the sparsity of available information in the documents. Our motivation for SROIE dataset was visual appearance discrepancy (ink color, additional hand written text, etc) and different content and/or layout in the receipts.
> >
> > ```
> >  - Authors say : "The reason we used FUNSD (Form Understanding in Noisy Scanned Documents) for sparsity is that this dataset is created with noisy data (e.g. originating from the disparity in scanning processes). So the shift between the source and target is the sparsity of available information in the documents. " . This statement makes little sense to this reviewer. How is noise related to text sparsity?  What is meant by "disparity in scanning processes"? How can scanning make sparser or denser?
> >
> > - The rationale for SROIE split is also highly intuitive. For a work like this, some sort of rationale (not intuitions) need to be provided on how the data is splitted.
> >
> >
> > ```
> > We also note that in our setting, there is no labeled data from the target domain, and hence there is no supervised training (which uses labels).
> > ```
> > Is not training using pseudo labels , a supervised learning  technically ? You are supervising using pseudo labels.
> >
> > ```
> > We have added a note about this in Sec 3.1 where we explain the inputs of the model.
> >
> > ```
> > The  question was what will be this vector, for a token for the question. The updated section 3.1 still does not say anything about it.
> >
> > ```
> > If it is masked then it is not included in the cross-entropy loss calculation because the attention mask for it is zero so the model essentially cannot make a prediction for it in that particular batch.
> > ```
> > In datasets like FUNSD, almost all the tokens have an associated tag or label. So if 15% are masked, it means 15% tokens are removed from the loss calculation. Would not it affect the learning. You are essentially telling the model that these tokens does not exist at all, when they are present in the image and they have a semantic tag associated with them. This reviewer believes that the effect of this on learning needs to be studied.

---

> > > ### Author Response · Authors · 2023-04-18
> > > **Continuing rebuttal discussion with Reviewer hWWC [1/2]**
> > >
> > > We would like to thank the reviewers for taking the time to read our response and engaging in the rebuttal discussion. Please see our comments below:
> > >
> > > `What authors highlight as differences compared to existing TTA are i) it is the only approach that combines multimodal self-supervised representation learning, However this reviewer would like to reiterate that this is nothing but MVLM that is been widely used in BERT-like models like LayoutLM, VLBERT, VILBERT etc. Secondly, using entropy to pick pseudo labels is a popular technique. More the confusion, higher the change of error is a general rule applied to pick pseudo labels. For example see paper https://openreview.net/pdf/c979bcaed90f2b14dbf27b5e90fdbb74407f161b.pdf , section 3.3.`
> > >
> > > Just to clarify, we do not propose that our paper is the first to propose an MVLM objective (please let us know if we are overclaiming the novelty aspects in our submission and we would be happy to fix it). As the Reviewer has noted, it has been used in many other works for multimodal self-supervised pre-training. However, none of those works consider the test-time adaptation setting, which brings unique challenges compared to pre-training.
> > >
> > > Regarding the mentioned reference, we have cited the mentioned paper (Rizve et al.) in our Introduction and Related Work sections and it differs from our Objective II in Sec 3.2 in two main ways: i) that they are doing semi-supervised learning on one distribution not test-time adaptation that involves adapting to a different distribution that training at test time and ii) that they propose to filter pseudo labels based on both confidence and uncertainty measured with Dropout (Eq 5 in their paper) but we argue uncertainty measured with entropy works better under distribution shifts.
> > >
> > > `Authors say : "The reason we used FUNSD (Form Understanding in Noisy Scanned Documents) for sparsity is that this dataset is created with noisy data (e.g. originating from the disparity in scanning processes). So the shift between the source and target is the sparsity of available information in the documents. " . This statement makes little sense to this reviewer. How is noise related to text sparsity? What is meant by "disparity in scanning processes"? How can scanning make sparser or denser?`
> > >
> > > Just to clarify, noise and sparsity are two different issues in FUNSD dataset. According to the FUNSD paper, the disparity in scanning processes comes from the scanning process of their old receipts from the 1980s-1990s where the scanned images have corruption and low resolution which we think makes OCR outputs have higher error causing noise in the input bounding boxes. On the other hand, they mention the sparsity issue which is due to the fact that the receipts are from different industries (marketing, science, advertisement, etc.) where they have an unbalanced set of labels provided per industry field, i.e., tax forms are filled with lots of information whereas the ones from tobacco companies are mostly sparsely filled. This creates a challenge for a model that is trained on dense forms from a certain category to make predictions on forms from a different category with new entities that don’t even have tags because when running OCR on the target domain with sparsely field data, we have a sequence of detected bounding boxes which cannot be mapped to any label and also the correlation that the model has leaned between items in the sequence on source domain does not apply in the target domain anymore. We have also added a note about this to Sec 4.1 for more clarification.
> > >
> > > `The rationale for SROIE split is also highly intuitive. For a work like this, some sort of rationale (not intuitions) need to be provided on how the data is splitted.`
> > >
> > > Creating datasets with visual disparity to study robustness and generalization is quite common [1]. For example in [1], they create algorithmic noise such as motion blurring, zoom blurring, adding gaussian noise, frosting, Fogging, etc. This is done to replicate the natural noise that exists in real-world datasets. Here, we do have a real-world dataset that contains certain visual noise that we leverage to split the data into two categories. While we agree this is not the only way to create an adaptation dataset, we believe it is a great starting point to create the first domain adaptation dataset for document data that can be used as a standard benchmark.
> > >
> > > [1] Hendrycks, Dan, and Thomas Dietterich. "Benchmarking Neural Network Robustness to Common Corruptions and Perturbations." International Conference on Learning Representations.

---

> > > > ### Author Response · Authors · 2023-04-18
> > > > **Continuing rebuttal discussion with Reviewer hWWC [2/2]**
> > > >
> > > > `Is not training using pseudo labels , a supervised learning technically ? You are supervising using pseudo labels.`
> > > >
> > > > The conventional terminology for supervised learning assumes that the learning happens using the ground truth labels that are provided by humans. Some works also consider this terminology when pseudo labels are used instead of labels collected from humans. To avoid confusion, we refrain from using the term “supervised” learning when describing our test-time adaptation method.
> > > >
> > > > `The question was what will be this vector, for a token for the question. The updated section 3.1 still does not say anything about it.`
> > > >
> > > > The note about this in Sec 3.1 is in the 6th line from the bottom of page 4 that says `Note that for the VQA task, text input sequence is also prepended with the question`. The whole input sequence passed to the tokenizer looks like the following in code if this helps clarifying better:
> > > > ```
> > > > encoding = tokenizer([questions], [words], [boxes], max_length=max_length, padding="max_length", truncation=True,return_tensors="pt")
> > > > ```
> > > > where the tokenizer is the LayoutLMv2Tokenizer from https://github.com/huggingface/transformers/blob/v4.28.1/src/transformers/models/layoutlmv2/tokenization_layoutlmv2.py#L206
> > > >
> > > > `In datasets like FUNSD, almost all the tokens have an associated tag or label. So if 15% are masked, it means 15% tokens are removed from the loss calculation. Would not it affect the learning. You are essentially telling the model that these tokens does not exist at all, when they are present in the image and they have a semantic tag associated with them. This reviewer believes that the effect of this on learning needs to be studied.`
> > > >
> > > > From an optimization standpoint, the model trains through getting penalized for wrong predictions such that by calculating the loss on those predictions, it finds out how much to adjust its parameters in order to get closer to the ground truth. When we don’t provide a token, we don’t make the model to predict anything either and hence no penalization takes place as a consequence. So this might not be ideal for learning the labels, it is not hurting it either. It is actually helping the model in a different way, learning the words in the target domain by predicting the randomly masked words, as a way of adjusting to the target data distribution.
> > > >
> > > > Our experiments show that this rule-of-thumb 15% masking empirically worked for representation learning and we did not tune this parameter (as we intended to minimize reliance of our method on hyperparameter tuning). Since masking is done randomly at each iteration (i.e. different 15% of the tokens are masked out), the model wouldn’t be biased to learn some tokens more or less. As a matter of fact there are recent studies that looked into this value for BERT and they argue it could still preserve 95% fine-tuning performance if it is pretrained with an extremely high masking ratio of 80% [1]. Overall, we agree that tuning the percentage value can bring some benefits, if hyperparameter tuning is an option.
> > > >
> > > > [1] Wettig, Alexander, et al. "Should you mask 15% in masked language modeling?." arXiv preprint arXiv:2202.08005 (2022).

---

> > > > > ### Comment · Reviewer_hWWC · 2023-04-19
> > > > > **hWWC's responses 2/2**
> > > > >
> > > > > ```
> > > > > The whole input sequence passed to the tokenizer  looks like the following
> > > > > encoding = tokenizer([questions], [words], [boxes], max_length=max_length, padding="max_length", truncation=True,return_tensors="pt")
> > > > >
> > > > >
> > > > > ```
> > > > > This reviewer understands that the input the model is formed by concatenating question tokens, OCR tokens  and the boxes or the visual tokens, The question is specifically about the 2D position embedding (shown in green boxes in the Figure 2.). For a question token, what will be the 2D position embedding. ? In my previous responses , requested the authors to present a figure that shows the schematic of the proposed approach for the VQA case. That would help to understand this better.
> > > > > The way the paper talks about the input the model is confusing to the readers. Authors could at least stick with the terminology used in LayoutLM paper that is used in this work as the base model .  What do "words" mean in the above code snippet? This reviewer believes authors meant to say "OCR tokens". Using "words" is misleading since questions too have words in it. And what authors call as "boxes" in the equation seems to be one quadrant from the image (called as v1, v2, v3, v4 in LayoutLM v2 paper). Calling them "boxes" is again misleading since the OCR tokens too have a bounding box associated with them. The v1,v2,v3,v4 are visual tokens in LayoutLM v2 parlance.
> > > > > So both the visual tokens and OCR tokens have a box associated with them. This reviewer's question is about the tokens from the question, what will be the 2D position embedding for the tokens from the question?
> > > > > This confusion can be avoided, if figures depicting the model training is shown for each of the tasks (FUNSD, SROIE and DocVQA) are shown separately in the supplementary.

---

> > > > > > ### Author Response · Authors · 2023-04-22
> > > > > > **Continuing rebuttal discussion with Reviewer hWWC [1/1]**
> > > > > >
> > > > > > Once again we thank the reviewer for taking the time to read our response and continue the discussion.  We highly appreciate your patience and dedication. Please find our answers below.
> > > > > >
> > > > > > `However this reviewer likes to point out that the the uncertainty aware pseudo label selection is not new.`
> > > > > >
> > > > > > We appreciate the note and agree with the point on the uncertainty-aware pseudo labeling not being novel by itself, but rather its application for DocTTA being novel. To prevent overclaiming, we have modified the sentence in the paper as "We introduce an uncertainty-aware per-batch pseudo labeling selection mechanism, which makes more accurate predictions compared to the commonly-used pseudo labeling techniques in CV". Please let us know if you think there is any other place where the novelty claim is not supported.
> > > > > >
> > > > > > `What authors wrote about why sparsity causes distribution shift is not convincing. This reviewer agrees that there is no known ways to split the dataset since the dataset was originally not collected for domain adaptation experiments. However, the current splitting mechanism seems to be highly intuitive. One possible way to explore this further is to try splitting in different ways. For example one can split FUNSD based on document source or the topic/industry of the document.`
> > > > > >
> > > > > > We note that there is no metadata available based on the document source or the topic/industry, so such splitting is highly nontrivial.
> > > > > >
> > > > > > Our splitting based on sparsity indeed creates distribution differences as different documents would have different likelihood of containing information for certain entities.  For example, there are 12% more sparse labels (entities with no label) in the target domain compared to source. Or 5% of the entities on source are from the “question” category whereas this number is ~2.5% in the target domain. Nonetheless, we observe that the shift is prominent also from the results between “Source-only” vs. domain adaptation and TTA methods, as the latter bring the majority of their benefits for significant distribution shifts.
> > > > > >
> > > > > >
> > > > > > `You are essentially asking a model to do form understanding when 15% tokens are removed from the context. My earlier comment highlighted this.`
> > > > > >
> > > > > > We acknowledge that the information is randomly being missed but at the same time the model is trying to predict those words, being supervised with the MVLM loss. Also we have different randomly selected 15% tokens masked at each iteration, so we do not think this constitutes a fundamental issue and also our experimental results back up this as DocTTA demonstrates significant improvements with this setting.
> > > > > >
> > > > > >
> > > > > >
> > > > > > `The question is specifically about the 2D position embedding (shown in green boxes in the Figure 2.). For a question token, what will be the 2D position embedding. ?`
> > > > > >
> > > > > > The token level 2D positional embedding or called layout embedding is the concatenation of six bounding box features in x and y directions as follows:
> > > > > > `Concat PosEmb2D_x(x_min, x_max, width), PosEmb2D_y(y_min, y_max, height)`
> > > > > > where x_min, x_max, width, y_min, y_max, height are obtained using the OCR outputs which we have illustrated in Figure 5 of Sec A.8 in the appendix.
> > > > > >
> > > > > > `What do "words" mean in the above code snippet? This reviewer believes authors meant to say "OCR tokens". Using "words" is misleading since questions too have words in it.`
> > > > > >
> > > > > > We refer to “OCR’s output text sequence” as words. Similar to the LayoutLMv2 paper, we also follow the common practice of using WordPiece to tokenize the OCR text sequence and assign each token to a certain segment $s_i \in {[A], [B]}$. Then, we add [CLS] at the beginning of the sequence and [SEP] at the end of each text segment. Extra [PAD] tokens are appended to the end so that the final sequence’s length is exactly the maximum sequence length L. The final word embedding is the sum of three embeddings: 1) Token embedding which represents the token itself, 2) 1D positional embedding which represents the token index, and 3) segment embedding which is used to distinguish different text segments.
> > > > > >
> > > > > > `what authors call as "boxes" in the equation seems to be one quadrant from the image (called as v1, v2, v3, v4 in LayoutLM v2 paper).`
> > > > > >
> > > > > > No, when OCR detects a word, it outputs its text and the bounding box around. We refer to that bounding box as a box which is defined using six values as illustrated in Figure 5.
> > > > > >
> > > > > > `what will be the 2D position embedding for the tokens from the question?`
> > > > > >
> > > > > > Questions are parsed as text and there is no bounding box associated with them and hence no 2D position embedding is generated for them because they are not outputs of the OCR - they are provided for each document and are tokenized the same way as we tokenize the “OCR’s output text sequence” as described above.

---

> > > > ### Comment · Reviewer_hWWC · 2023-04-19
> > > > **hWWC's responses  1/2**
> > > >
> > > > - This reviewer acknowledges the fact that authors use MVLM for test time domain adaptation not in a typical supervised setting. This reviewer wanted to point out (for the sake of other readers) that MVLM, since it is a self supervised technique can be trivially extended to test time adaptation settings since MVLM doesn require extra labelling.
> > > >
> > > > -  The paper by Rizve et al. was referred to show that picking predictions with low uncertainity as pseudo labels is a  popular technique. High confident predictions, with low entropy or uncertainty is  a straightforward technique to pick pseudo labels. Following are some recent papers that use this. Note that these were found from a quick google search. May not be the popular ones
> > > >
> > > >    1. https://proceedings.mlr.press/v180/huang22a/huang22a.pdf
> > > >    2. https://openaccess.thecvf.com/content/CVPR2021/papers/Hu_SimPLE_Similar_Pseudo_Label_Exploitation_for_Semi-Supervised_Classification_CVPR_2021_paper.pdf
> > > >
> > > > What authors use for picking pseudo labelling is exactly same as this existing, popular technique. This is noted , not as a criticism, but to highlight the fact, so that this can be considered for the papers evaluation. For example this sentence in the paper "We introduce a new uncertainty-aware per-batch pseudo labeling selection mechanism, which makes more accurate predictions compared to the commonly-used pseudo labeling techniques in CV" says that the pseudo label selection mechanism is  new. However this reviewer likes to point out that the the  uncertainty aware pseudo label selection is not new.
> > > >
> > > > ```
> > > > Just to clarify, noise and sparsity are two different issues in FUNSD dataset
> > > > ```
> > > > Quoting what authors said in the first response: "The reason we used FUNSD (Form Understanding in Noisy Scanned Documents) for sparsity is that this dataset is created with noisy data (e.g. originating from the disparity in scanning processes). So the shift between the source and target is the sparsity of available information in the documents." Request the authors to read this. This response clearly states that authors  picked FUNSD for sparsity since it is created with noisy data. What authors wrote about why sparsity causes distribution shift is not convincing. This reviewer agrees that there is no known ways to split the dataset since the dataset was originally not collected for domain adaptation experiments.  However, the current splitting mechanism seems to be highly intuitive. One possible way to explore this further is to try splitting in different ways. For example one can split FUNSD based on document source or the topic/industry of the document.
> > > >
> > > >
> > > >
> > > >
> > > >
> > > > ```
> > > > When we don’t provide a token, we don’t make the model to predict anything either and hence no penalization takes place as a consequence. So this might not be ideal for learning the labels, it is not hurting it either. It is actually helping the model in a different way, learning the words in the target domain by predicting the randomly masked words, as a way of adjusting to the target data distribution.
> > > > ```
> > > > This reviewer was not talking about the MVLM objective, but the objective for the downstream that uses the pseudo labels. For example, in FUNSD, this is about assigning semantic labels or tags to the tokens. When 15\% of the tokens are masked as part of the MVLM, for the other task, i.e, the sequence labelling task for FUNSD, 15% tokens are missing. The model is made to label the tokens with  partial information.  Consider the example given in Figure 2 in FUNSD paper (https://arxiv.org/pdf/1905.13538v2.pdf). Imagine the tokens "usd", "spring" and "festival" are masked, and you are trying to assign a label (or semantic entity tag) to the word "the". This technically makes the model to assign a label without the full context. This is happening since masking is essential for MVLM, but not required (or detrimental) for the other objective. You are essentially asking a model to do form understanding when 15\% tokens are removed from the context. My earlier comment highlighted this.

---

### Author Response · Authors · 2023-04-15
**Response to all reviewers**

We would like to thank all the reviewers for their constructive and comprehensive feedback that has helped us improving our paper. We have addressed all the reviewers' comments by replying to them individually. We have also updated the revised version of the paper based on all the feedbacks and the requested changes. We hope we have been able to clarify your great points and we're happy to answer any further questions or concerns.

---

### Decision · Action_Editors · 2023-05-27

**Recommendation:** Accept as is

**Comment:**

The reviewers acknowledged that the approach, which is based on self-supervised learning (via masked visual-language modeling) on the source domain followed by pseudo-labeling on the target domain, is reasonable and well-supported by convincing empirical evidence. One reviewer expressed concern that both self-supervised learning and pseudo-labeling have been extensively studied in the literature. The authors clarified that while these methods are well-established in the literature, their usage in the context of test-time adaptation for VDU is novel. The reviewers generally found that the overall contribution of this paper, which also includes a new benchmark, outweighs this concern, and they recommended acceptance.

This AE agrees with the reviewers' assessment and recommendation for acceptance. The novel application of self-supervised learning and pseudo-labeling in the context of test-time adaptation adds value to the existing literature, and the new benchmark addresses an important research problem. Therefore, based on the reviewers' positive evaluations and the AE's own evaluation, the paper is recommended for acceptance.


**Audience:**

As all three reviewers agreed, this paper addresses an interesting problem in document understanding. The proposed approach is well-supported by compelling empirical evidence, and the benchmark could serve as a valuable resource for the community. One reviewer specifically mentioned that this paper could be highly beneficial for those seeking to enter the field.








**Claims And Evidence:**

The paper proposes test-time domain adaptation for visual document understanding (VDU) tasks. The reviewers all agreed that the paper tackles an interesting and important problem. The proposed approach is deemed reasonable, and the experimental results are convincing. They also acknowledged the significant benchmark contribution of this paper, which builds upon existing datasets for domain adaptation. The writing quality, including the extensive details on the benchmark construction, was unanimously praised by the reviewers.